# Revealing proteome-level functional redundancy in the human gut microbiome using ultra-deep metaproteomics

Leyuan Li [1,2,7], Tong Wang[3,7], Zhibin Ning[2], Xu Zhang [2], James Butcher [4], Joeselle M. Serrana[2], Caitlin M. A. Simopoulos [2], Janice Mayne[2], Alain Stintzi [4], David R. Mack[5], Yang-Yu Liu [3,6] ✉ & Daniel Figeys [2] ✉

Functional redundancy is a key ecosystem property representing the fact that different taxa contribute to an ecosystem in similar ways through the expression of redundant functions. The redundancy of potential functions (or genome-level functional redundancy $FR_g$) of human microbiomes has been recently quantified using metagenomics data. Yet, the redundancy of expressed functions in the human microbiome has never been quantitatively explored. Here, we present an approach to quantify the proteome-level functional redundancy $FR_p$ in the human gut microbiome using metaproteomics. Ultra-deep metaproteomics reveals high proteome-level functional redundancy and high nestedness in the human gut proteomic content networks (i.e., the bipartite graphs connecting taxa to functions). We find that the nested topology of proteomic content networks and relatively small functional distances between proteomes of certain pairs of taxa together contribute to high $FR_p$ in the human gut microbiome. As a metric comprehensively incorporating the factors of presence/absence of each function, protein abundances of each function and biomass of each taxon, $FR_p$ outcompetes diversity indices in detecting significant microbiome responses to environmental factors, including individuality, biogeography, xenobiotics, and disease. We show that gut inflammation and exposure to specific xenobiotics can significantly diminish the $FR_p$ with no significant change in taxonomic diversity.

The human gut microbiome is a complex ecosystem harboring trillions of microorganisms. Its taxonomic composition, functional activity and ecosystem processes have important consequences on human health and disease. It is crucial to study the human gut microbiome in the context of ecological communities[1]. The most frequently applied ecosystem measures of the gut microbiome have been taxonomic diversity calculators[2], which are not sufficiently informative in assessing the functional states of the gut microbial

[1]State Key Laboratory of Proteomics, Beijing Proteome Research Center, National Center for Protein Sciences (Beijing), Beijing Institute of Lifeomics, 102206 Beijing, China. [2]School of Pharmaceutical Sciences and Ottawa Institute of Systems Biology, Faculty of Medicine, University of Ottawa, Ottawa, ON K1H 8M5, Canada. [3]Channing Division of Network Medicine, Department of Medicine, Brigham and Women's Hospital and Harvard Medical School, Boston, MA 02115, USA. [4]Department of Biochemistry, Microbiology and Immunology, Faculty of Medicine, University of Ottawa, Ottawa, ON K1H 8M5, Canada. [5]Department of Paediatrics, Faculty of Medicine, University of Ottawa and Children's Hospital of Eastern Ontario Inflammatory Bowel Disease Centre and Research Institute, Ottawa, ON K1H 8L1, Canada. [6]Center for Artificial Intelligence and Modeling, The Carl R. Woese Institute for Genomic Biology, University of Illinois at Urbana-Champaign, Urbana, IL 61801, USA. [7]These authors contributed equally: Leyuan Li, Tong Wang. ✉e-mail: yyl@channing.harvard.edu; dfigeys@uottawa.ca

ecosystem. Moreover, diversity indices have widely varied between studies with some studies reporting lower diversity indices in disease microbiomes than in healthy controls[3], while other studies reported higher diversity indices in disease versus controls[4,5]. Numerous examples including those two are raising the concern of correlating diversity values to health and disease status[6]. Indeed, in a microbial ecosystem, diversity is just one of the properties which does not necessarily correlate with its functionality. Much more valuable and informative insight should be gained by considering the microbiome from a functional perspective.

Functional redundancy (FR) is a key property of ecosystems[7]. FR describes the ability of multiple taxonomically distinct organisms to contribute in similar ways to an ecosystem through having redundant functional traits[8-10]. A high level of FR implies that members in a community may be substitutable with little impact on the overall ecosystem functionality[8]. In the human gut microbiome, it has been found that FR is a common event. For example, dietary carbohydrates can be processed by either *Prevotella* (from the phylum Bacteroidetes) or *Ruminococcus* (from the phylum Firmicutes), and short-chain fatty acids can be produced by multiple predominant genera: *Phascolarctobacterium*, *Roseburia*, *Bacteroides*, *Blautia*, *Faecalibacterium*, *Clostridium*, *Subdoligranulum*, *Ruminococcus* and *Coprococcus*. Moreover, large-scale, consortium-driven metagenomic projects such as the Human Microbiome Project (HMP) have found that, regardless of the body site, within a healthy population the carriage of microbial taxa varies tremendously, while the gene compositions or functional profiles remain remarkably stable[2]. This is a strong signal of FR in the human microbiomes.

We emphasize that most of the previous studies on FR of human microbiomes are very conceptual, rather than quantitative. Recently, a computational pipeline was developed to quantify the genome-level FR of microbiome samples from whole-metagenome shotgun sequencing data[11]. This pipeline was based on the genomic content network (GCN), a bipartite graph in which taxa are connected to the genes in their genomes. Importantly, this GCN-based FR calculation was derived without any regard for whether these genes are expressed. In other words, the FR calculated from the GCN only represents the redundancy of potential functions of a microbiome sample (i.e., the genome-level FR), rather than the redundancy of expressed functions (e.g., the proteome-level FR).

In this work, we present a pipeline to quantify the proteome-level FR of microbiome samples from deep metaproteomics data. Metaproteomics is a powerful tool that measures expressed proteins in a microbiome based on liquid chromatography-tandem mass spectrometry (LC-MS/MS) techniques[12-14]. Our pipeline is based on the construction of the proteomic content network (PCN) for each microbiome sample by linking the taxa to their proteins. For each PCN, its proteome-level FR ($FR_p$) is defined as the part of its taxonomic diversity that cannot be explained by its functional diversity. $FR_p$ is an informative metric carrying contributions from the presence/absence of the functions, protein abundances of the functions and biomasses of the proteomes. We demonstrate that $FR_p$ sensitively responds to environmental factors affecting the microbiome.

## Results

### Construction of proteomic content networks from an ultra-deep metaproteomic approach

We define the proteomic content network (PCN) of a microbiome sample as a bipartite graph connecting each microbial taxon to all expressed functions from the taxon's proteome. In order to gain the deepest possible understanding of sample-specific PCN of the human gut microbiome, we first developed an ultra-deep metaproteomics approach based on high-pH reversed-phase fractionation[15] and high-

resolution LC-MS/MS analysis (Fig. 1a–e and *Methods*). Briefly, aliquots from gut microbiome samples were subjected to protein extraction and digestion, followed by being separated into 48 fractions at a 1 min interval and pooled at an interval of 12 wells. Using this workflow, we analyzed samples from four individuals' ascending colons (Supplementary Table S1). Database searches of the resulting 12 LC-MS/MS RAW files of each individual microbiome sample were performed using the MetaPro-IQ approach[16]. We first used the Integrated Gene Catalog (IGC) database of the human gut microbiome[17] to perform the database search. On average, 69,280 unique peptides were identified, and 30,686 protein groups were quantified per microbiome sample. Using a "protein-peptide bridge" method (Fig. 1f and Supplementary Note S1), functions that were annotated by protein groups and taxonomy that were identified by unique peptides were linked to constructing the sample-specific PCN (Fig. 1g, h). Since some proteins or peptide sequences are shared between two or more organisms in complex microbial communities, as a trade-off between taxonomic resolution and protein coverage, we computed the PCN on the genus level. Protein-level biomass contributions of each genus is represented by its summed unique peptide abundances (Fig. 1i). In terms of functional annotations, the Kyoto Encyclopedia of Genes and Genomes (KEGG) database has been widely used in functional metrics such as the genomic-level functional redundancy[11]. However, it is common in metaproteomic studies that a certain proportion of proteins does not have a KEGG annotation. Indeed, in this dataset, there were a total of 50,216 protein groups identified, among which, 46,095 (91.7%) were successfully annotated with clusters of orthologous groups (COGs), while only 37,795 (75.3%) were annotated with KEGG KOs. Therefore, we complemented the KEGG annotations with COG to achieve better coverage (denoted as KEGG-COG annotation; see Supplementary Note S2 for more comparisons). This annotation will be applicable to metaproteomic-based functional redundancy computations without the need for the samples' metagenomes.

In addition, to facilitate direct comparisons of redundancy or network metrics between the GCN and the PCN, we further used the samples' paired metagenomes to generate Prodigal-predicted protein sequences as the database to perform another metaproteomic database search. An average of 65,541 unique peptides and 29,392 protein groups per sample were obtained from the search. The Prodigal sequences were blasted against the UHGP database for taxonomic matches. In addition, KEGG-COG annotations were performed. GCNs or PCNs were then computed by summing read counts or protein intensities at each taxon-function incidence (see *Methods*).

### Computation of within-sample proteome-level functional redundancy

We define proteome-level functional redundancy ($FR_p$) of a microbiome sample as the part of the alpha diversity of taxonomic protein biomass contributions ($TD_p$) that cannot be explained by the alpha functional diversity ($FD_p$) (Supplementary Fig. S1a):

$$FR_p \equiv TD_p - FD_p = \sum_{i=1}^{S} \sum_{j \neq i}^{S} \left(1 - d_{ij}\right) p_i p_j \qquad (1)$$

where $S$ is the number of taxa in the sample, and $p_i$ is the proportion of protein-level biomass of taxon $i$ within the sample. The protein-level biomass in the PCN can be approximated by summing up the intensities of unique peptides in each taxon, which has been previously shown to be a good representation of microbiome diversity[18]. $d_{ij}$ denotes the functional distance between taxa $i$ and $j$ measured by the weighted Jaccard distance between their proteomic contents (see *Methods*), $TD_p = \sum_{i=1}^{S} \sum_{j \neq i}^{S} p_i p_j = 1 - \sum_{i=1}^{S} p_i^2$ is the alpha taxonomic diversity measured by the Gini-Simpson index, and

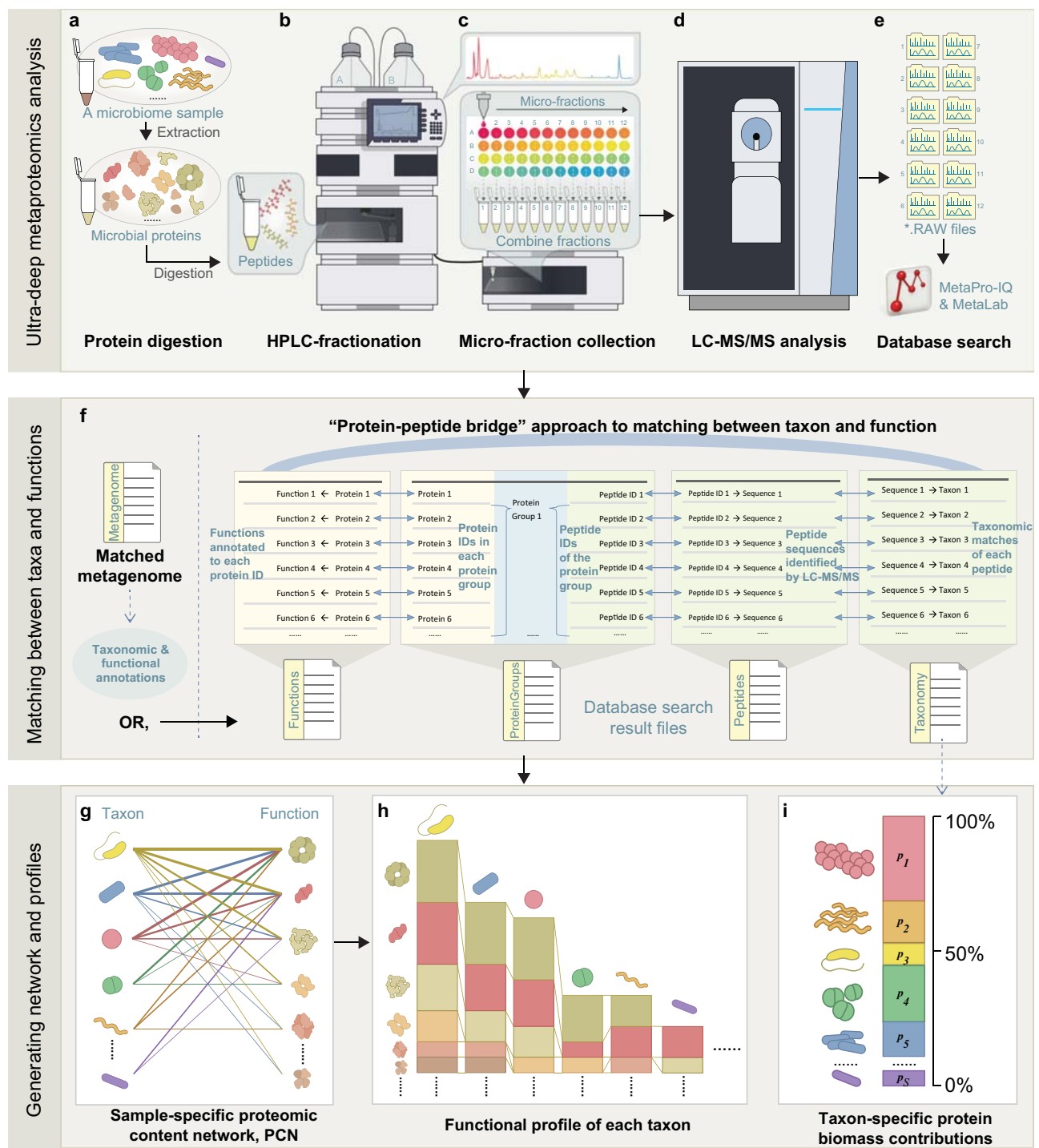

**Fig. 1 | Computation of within-sample proteomic-content network and functional, taxonomic profiles using an ultra-deep metaproteomics approach.**
**a** Each individual's gut microbiome sample was subjected to protein extraction. Then, purified proteins were digested by trypsin. **b** The resulting peptides were fractionated using a high-pH reversed-phase approach. **c** 48 micro-fractions were combined into 12 samples prior to LC-MS/MS analysis. **d** In this study, LC-MS/MS analysis was performed using Exploris 480 with a 1 h gradient for each fraction. **e** The LC-MS/MS *.RAW files were searched against a protein database using MetaPro-IQ workflow and MetaLab. **f** When matched metagenomes are available,

taxonomic and functional annotations of the metagenomes can be used for PCN generation. Alternatively, a protein-peptide bridge approach can be used for generating the PCNs from MetaLab result tables (see details in *Methods* and Supplementary Notes 1) without the need of matched metagenomes. **g** The sample-specific PCN is a bipartite plot linking taxon to their expressed functions, with the links weighed by protein abundances of each taxon. **h** The PCN can also be converted as functional profiles of each taxon. **i** The taxon-specific protein biomass contributions (percentage) were calculated from the taxonomy table based on summing up unique peptides of each taxon.

$\mathrm{FD}_p = \sum_{i=1}^{S} \sum_{j \neq i}^{S} d_{ij} p_i p_j$ is the alpha functional diversity measured by Rao's quadratic entropy. Notably, in contrast to computing a genome-level functional redundancy for which the $d_{ij}$ values can be computed from a reference GCN of all samples in the dataset, for

PCN, since protein expression levels can sensitively respond to environmental changes, within-sample proteomes should be used. We illustrate why it is impossible to construct and use a reference PCN in Supplementary Fig. S2.

By normalizing the $FR_p$ with the microbiome's alpha taxonomic diversity, we have:

$$nFR_p = \frac{FR_p}{TD_p} \qquad (2)$$

We highlight that $FR_p$ and $nFR_p$ are complex metrics that are derived from several key components and properties, i.e., functional abundances, taxonomic proteome-level biomass, network topology, and functional distances between different taxa. Using simple conceptual communities, we showed that when members of a community express the same list and abundances of functions, $nFR_p$ equals to 1 (Supplementary Fig. S1b); while when members of a community have totally different list of functions, $nFR_p$ equals to 0 (Supplementary Fig. S1c). Other than the two extreme cases, the value $nFR_p$ falls between 0 and 1 (Supplementary Fig. S1d). We emphasize that $nFR_p$ of a microbiome sample can be interpreted as the average functional similarity (or overlap) of two randomly chosen members in the sample.

We demonstrate the sensitivity of $FR_p$ and $nFR_p$ using in silico communities generated with genomes and proteomes of single bacterial strains (i.e., *Phocaeicola* (*Bacteroides*) *vulgatus* ATCC 8482, *Bacteroides ovatus* ATCC 8483, *Bacteroides uniformis* ATCC 8492, *Blautia hydrogenotrophica* DSM 10507, *Escherichia coli* DSM 101114). Proteomes of these strains cultured in four different media (basal medium with or without added glucose, sucrose or kestose) were obtained from our previous study[19] (Supplementary Fig. S3a). We first used the genomes and proteomes (in basal media) to generate different three-member communities in silico (Supplementary Fig. S3b and S3c). When all three members belong to *Bacteroides* or *Phocaeicola* genera, the community's genome- and proteome-level functional redundancy were both higher compared with the other combinations. The redundancies decreased as the community becomes more diverse on the genus level. In Supplementary Fig. S3b and S3c, despite genome-level functional redundancy may seem predictive of the proteome-level functional redundancy, we emphasize that, in principle, genome-level functional redundancy only responds to the change of microbial abundances. When we further replaced the proteomes of strains with the ones cultured in the presence of different sugars (and maintained microbial abundances), the levels of $FR_p$ and $nFR_p$ showed fluctuations (Supplementary Fig. S3d). This suggests that $FR_p$ and $nFR_p$ are sensitive to a community's functional responses, even induced solely by proteome alterations while microbial abundances are unchanged.

### Exploration of PCN topology and its ecological implication

We investigated the topological properties of these PCNs, and compared them with their corresponding GCNs obtained from shotgun metagenomics. The PCNs of metagenome and IGC databases-based search yielded similar depth, both achieved reasonable depths compared with each individual's respective GCNs (Supplementary Fig. S4). Figure 2a shows a tripartite plot connecting microbial phyla and functional categories annotated according to genes and proteins from one individual microbiome (HM454, IGC database-based search with COG annotations). This demonstrated that while some functional categories (e.g., energy production and conversion (C), carbohydrate metabolism and transport (G) etc.) showed expression from predicted functions in most phyla, some genes linked to functions such as RNA processing and modification (A) and mobilome (X) that were rarely expressed from the genes. Similar results were found for other samples (Supplementary Figs. S5–7). We wondered what is the ecological implication of such selective functional expression from the GCN to the PCN. To investigate this, we explored the community assembly results in silico through a model framework[20] that incorporates cross-feeding interactions into the classical MacArthur's Consumer-Resource Model[21] (see *Methods* and Fig. 2b–d). We hypothesized that the topological property of the GCN and PCN such as connectance (i.e.,

the number of links divided by the number of maximal possible links) strongly influence the assembly results. Therefore, we modeled system dynamics under variable environmental conditions and investigated how the difference in the connectance between the GCN ($0.22 \pm 0.02$; Mean $\pm$ SD, $N = 4$) and the PCN ($0.05 \pm 0.02$; Mean $\pm$ SD, $N = 4$; metagenome database-based search) affects the richness of assembled communities. Simulations of system dynamics show that the model's resource consumption matrix C following the PCN's connectance always maintains higher microbiome richness and survival of more diverse species than following the GCN's connectance, despite variations in dilution rates (Fig. 2e–g), byproduct fractions (Supplementary Fig. S8a–c), and externally supplied nutrient diversity (Supplementary Fig. S8d–f), as well as different ratios between initial species abundances and initial metabolite/resource concentrations (Supplementary Fig. S9).

The intuitive relationship between biodiversity and functional redundancy in a community has been difficult to be quantified[22]. We sought to explore this relationship by examining the network topology. By visualizing incidence matrices of these PCNs, we observe highly nested structures (Fig. 3a and Supplementary Fig. S10) and found that the Nestedness metric based on Overlap and Decreasing Fill (NODF) were high in the PCNs (NODF = $0.28 \pm 0.01$; Mean $\pm$ SD, $N = 4$, metagenome database-based search), which are close to those of the respective GCNs (NODF = $0.36 \pm 0.05$; Mean $\pm$ SD, $N = 4$). Similarly, the PCNs based on IGC database-based search also resulted in high NODF values ($0.34 \pm 0.01$; Mean $\pm$ SD, $N = 4$). We then calculated the degree distributions of genera and functions in the PCNs and the GCNs, respectively. On the functional dimension, similar to previous observations in GCNs[11], the degree distributions of functions in both the GCN and PCN have fat tails, represented by some functions being associated with a high number of taxa (Fig. 3b and Supplementary Fig. S10). Similar nested topology and functional degree distributions can be observed in the PCNs generated with the IGC-based search (Supplementary Fig. S11). The high nestedness and fat-tail degree distribution of the PCN together suggests that specialist taxa are playing functional roles that are a subset of active functions from generalist taxa, which further indicates the existence of high redundancy of expressed functions in the human gut microbiome.

### Human gut microbiome has high protein-level FR

Since a potential function of any member in the GCN of the microbiome sample may or may not be expressed under a certain environmental condition, we anticipate that the proteome-level FR (i.e., $FR_p$ or $nFR_p$) of any microbiome sample should be no greater than its genome-level FR (i.e., $FR_g$ or $nFR_g$). Indeed, as shown in Fig. 4a, b, for the four individual microbiomes, we found that $FR_p$ (or $nFR_p$) preserved a high level of the $FR_g$ (or $nFR_g$; no significant difference by paired *t* test) as a result of maintenance of the difference between the corresponding TD and FD values (Fig. 4c, d).

How does the topology of the PCN contribute to the observed functional redundancy? To answer this question, we created four null networks by randomizing natural PCNs to change the network's inherent structural properties such as degree and nestedness. Considering the fact that PCNs are expressed from the GCNs, we randomized the PCNs under the background of each corresponding GCN (i.e., all edges on a randomized PCN are only selected from the edges of the sample's GCN). For the four null PCNs, Null-1 PCN is a completely randomized network under the GCN background, Null-2 PCN preserves the taxon degrees, Null-3 PCN preserves the function degrees, and Null-4 PCN preserves both taxon and function degrees (Fig. 4e). Each of the four null networks was randomized 10 times to validate the reproducibility of the results. We show that all four null networks had significantly lower $FR_p$ and $nFR_p$ compared to those of the natural PCNs (Fig. 4f). Figure 4f also suggests that both taxon and function degrees contributed to FR, as for each individual, there's an increasing

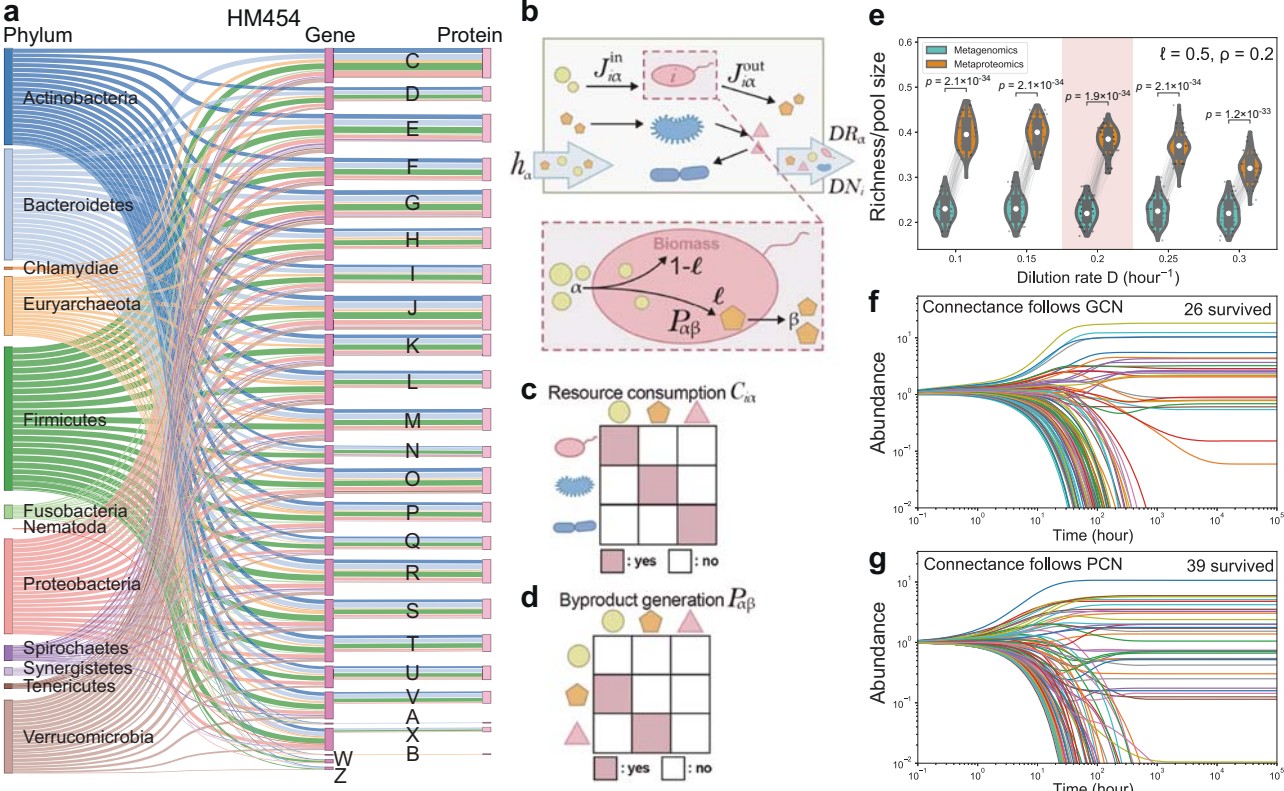

**Fig. 2 | Topology of proteomic content networks (PCNs) implies high richness.**
**a** A tripartite plot derived from GCN and PCN in microbiome HM454. Letters represent different functional categories in the Clusters of Orthologous Groups (COGs) database. Results of the other microbiomes are shown in Supplementary Figs. S5–S7. **b** Schematic of Consumer-Resource Model with cross-feeding. The community is assumed with nutrient supply $h_\alpha$ and dilution rate of resources $DR_\alpha$ and microbes $DN_i$ for each resource $\alpha$ and taxon-$i$. Each taxon-$i$ converts incoming consumption flux $J_{i\alpha}^{in}$ into byproduct flux $J_{i\alpha}^{out}$. A fraction $l \in [0,1]$ of the incoming resource is converted into byproduct $\beta$, and fraction $(1 - l)$ is assimilated into the biomass of taxon-$i$. **c** Resource consumption matrix $C_{i\alpha}$ describing the ability of taxon-$i$ in consuming resource $\alpha$. **d** Byproduct generation matrix $P_{\alpha\beta}$ describing conversion of a resource to a byproduct. Columns are consumed resources and rows are byproducts. **e**–**g** Effect of the variation of dilution rate on microbiome richness was simulated with consumption matrices with the same connectance as

the GCN ($C^{GCN}$) or PCN ($C^{PCN}$). The one pair of simulations (for GCN and PCN) in (**e**) were shown as (**f**, **g**). The default is dilution rate $D = 0.2 hour^{-1}$, byproduct fraction $l = 0.5$, and when 20 out of 100 nutrients are externally supplied ($\rho = 0.2$). Parameters for all panels are as follows: (**e**) $l = 0.5$ and $\rho = 0.2$; (**f**, **g**) $D = 0.2 hour^{-1}$, $l = 0.5$, and $\rho = 0.2$. Scattered dots and lines linking pairs of dots in (**e**) indicate each simulation paired between $C^{GCN}$ and $C^{PCN}$. Middle white dot in the box plot denotes median, the lower and upper hinges correspond to the first and third quartiles, the black line ranges from the $1.5 \times$ (interquartile range) below the lower hinge to $1.5 \times$ IQR above the upper hinge, and whiskers represent the maximum and minimum, excluding outliers. **** indicate statistical significance at the $p < 0.0001$ levels by two-sided Mann–Whitney-Wilcoxon U Test with Bonferroni correction. $N = 100$ times of independent simulations. More simulations by altering other factors are shown in Supplementary Figs. S8–9.

pattern of FR and nFR values from Null-1 to Null-4 PCNs. However, both taxon and function degrees are the same as the natural PCN in Null-4 PCN. Notably, there was a 18% decrease of nestedness in Null-4 PCNs ($NODF_{Null-4} = 0.23 \pm 0.03$; Mean $\pm$ SD, $N = 40$) in comparison to the natural PCN; in contrast, the corresponding decrease of $FR_p$ was $55\% \pm 11\%$ (Mean $\pm$ SD, $N = 4$), indicating that the network topology component of $FR_p$ does not fully explain the high functional redundancy. The functional relationships between specific pairs of taxa are also an important factor that shapes the high $FR_p$ in human gut microbiomes. We observed that all four null PCNs had altered distribution of $d_{ij}$ values from the natural PCNs (Fig. 4g).

**Protein-level FR outcompetes diversity indices in detecting microbiome responses to environmental factors**
We next examined whether different metaproteomic approaches could affect the network properties of gut microbiomes' PCNs and values of $FR_p$. Routine metaproteomic analyses are often performed without fractionation. In addition, samples are analyzed with different analytical protocols, different parameters and using different models of LC-MS/MS platforms, etc. Therefore, we evaluated four previously published datasets, briefly referred to as SISPROT[23], RapidAIM[24],

Berberine[25] and IBD[26] datasets, which vary considerably in the experimental protocol/platform and in the types of environmental factors (xenobiotics, biogeography, diseases status etc.) being interrogated (see details in Supplementary Tables S1–S3). It was notable that identification depths of these four datasets vary markedly, from 5612 protein groups and 4345 peptides per sample (Berberine) to 20,558 protein groups and 44,955 unique peptides per sample (SISPROT) (Supplementary Table S4). We found that PCNs in all the four datasets displayed highly similar topological structures with our new deep metaproteomics dataset, i.e., highly nested structure, and heterogeneous degree distributions of both taxa and functions (Supplementary Figs. S11 versus S12).

Given that topological structures of PCN appeared to be universal across platforms, we believe that $FR_p$ is applicable to different metaproteomic datasets. Indeed, we found that $nFR_p$ of the four datasets were comparable to our four ultra-deep metaproteomics samples (Fig. 5a, b, f–h versus Fig. 4b). In patients diagnosed with inflammatory bowel disease (IBD), $nFR_p$ levels were significantly lower than that of the non-IBD control individuals. There was no significant difference between the two different IBD subtypes Crohn's disease (CD) and ulcerative colitis (UC) (Fig. 5a). A significant decrease in $nFR_p$ was

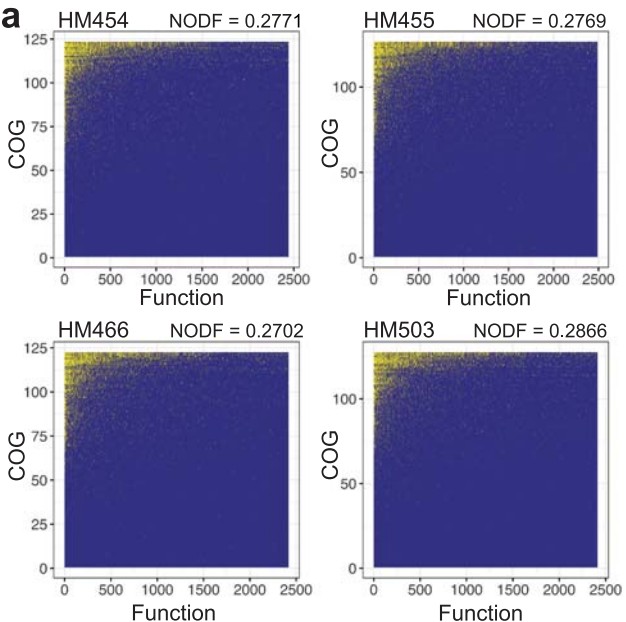

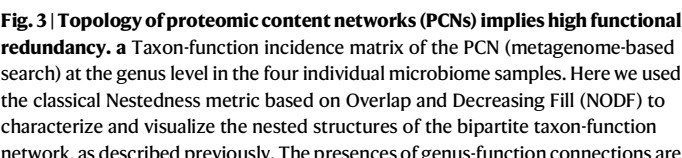

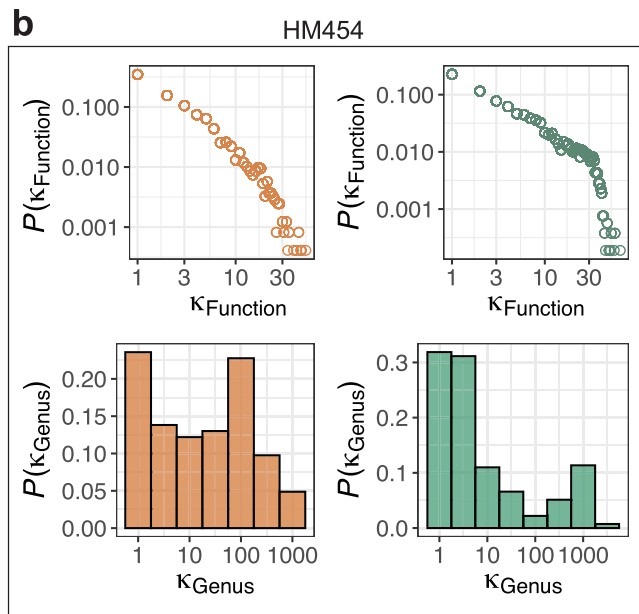

**Fig. 3 | Topology of proteomic content networks (PCNs) implies high functional redundancy. a** Taxon-function incidence matrix of the PCN (metagenome-based search) at the genus level in the four individual microbiome samples. Here we used the classical Nestedness metric based on Overlap and Decreasing Fill (NODF) to characterize and visualize the nested structures of the bipartite taxon-function network, as described previously. The presences of genus-function connections are shown in yellow points. **b** The unweighted degree distribution of functions in the PCN (upper left panel), that of genera in the PCN (lower left panel), that of functions in the GCN (upper right panel), and that of genera in the GCN (lower right panel) in microbiome HM454. Similar results of the other three individual microbiomes are shown in Supplementary Fig. S10. Source data are provided as a Source Data file.

found in inflamed regions from the terminal ileum (Fig. 5b), the diagnosis factor, as well as the inflammation factor which was nested in the biogeography factor significantly contributed to variance in $nFR_p$ (Supplementary Table S5). We next selected to compare the TI region between CD and the control microbiomes, as inflammation is likely to develop in the terminal ileum of CD patients[27]. $nFR_p$ significantly differentiated between inflamed and uninflamed regions (Fig. 5c, Wilcoxon rank-sum test, $p < 0.0001$), while $TD_p$ showed no significant difference between the two classes (Fig. 5d), and $FD_p$ showed significant difference but at the $p < 0.01$ level (Fig. 5e).

Inter-individual differences in $nFR_p$ levels were observed in the SISPROT, RapidAIM and Berberine datasets (Fig. 5f–h). Similarly, we detected significant differences in $TD_p$ and $FD_p$ between individual microbiomes (Wilcoxon rank-sum test; Fig. 5i, j). In the RapidAIM and Berberine datasets, a few xenobiotic compounds reduced $nFR_p$ levels (Fig. 5k and Supplementary Fig. S13a). Among these, the antibiotic rifaximin showed the most impact on the individual microbiomes with $nFR_p$ values decreased by $21.2 \pm 10.7\%$ (Mean $\pm$ SD, $N = 5$). Two-way ANOVA suggested that both the inter-individual variation and effect of compounds significantly contributed to $nFR_p$ variance (Supplementary Tables S6–S7). In contrast, $TD_p$ and $FD_p$ values were not as sensitive in detecting significant responses to drug treatments (Fig. 5l,m and Supplementary Fig. S13b,c). These together suggest that $nFR_p$ outcompetes diversity indices $TD_p$ and $FD_p$ in sensitively detecting microbiome responses to environmental factors.

Similar to the $nFR_p$ results, significant differences in NODF values were observed (Supplementary Fig. S14) in agreement with the within-sample $nFR_p$, which further suggests that a nested topological structure contributes to the values of $nFR_p$ in a microbiome sample.

### Alteration of between-proteome functional distances in disease and compound-treated microbiomes

To further find out the contribution of functional distances behind the response of within-sample $nFR_p$ to environmental factors, we examined the functional distance $d_{ij}$ between proteomes in each individual microbiome sample of the datasets. For the IBD dataset, PERMANOVA test showed that $d_{ij}$ values differed significantly between diagnosed patients (especially inflamed regions) and the non-IBD controls (Supplementary Table S8). Overall, the $d_{ij}$ distributions in both UC and CD samples showed a rightward shift from the control samples (Fig. 6a). Moreover, there was a rightward shift of the $d_{ij}$ distribution from healthy to inflamed gut regions (Fig. 6b). The volcano plot further showed that most of the $d_{ij}$ values were increased in the presence of inflammation (Fig. 6c), suggesting a significant contribution of between-proteome functional distance increase to the overall decrease of $nFR_p$ in IBD samples. We visualized the $d_{ij}$ values of genus pairs that significantly (at the $p < 0.001$ level by Wilcoxon rank-sum test) altered in inflammation using a heatmap coupled with hierarchical clustering (Supplementary Fig. S15a). Cluster 1 samples, which were mostly from inflamed regions of patients diagnosed with UC or CD, showed an overall increase of $d_{ij}$ values between proteome pairs. We showed that Cluster 1 was significantly higher in $nFR_p$, despite both its TD and FD were significantly lower than those in Cluster 2 (Supplementary Fig. S15b).

In contrast to the disease-induced alteration of $d_{ij}$ distributions, we also visualized the response of $d_{ij}$ values to xenobiotics and discovered that overall the $d_{ij}$ values have strong individual signatures and were significantly affected by the xenobiotic factor (Fig. 6d and Supplementary Table S9). We quantified the Jensen-Shannon (J-S) divergence of $d_{ij}$ distributions between drug treatments and the DMSO control (Fig. 6e). These results showed that ciprofloxacin, berberine, rifaximin, FOS, metronidazole, isoniazid, diclofenac and flucytosine significantly increased J-S divergence with the DMSO when compared to most other compounds (Fig. 6f). This was in agreement with our previous findings that these compounds (except flucytosine) resulted in global alterations in individual microbiome functionality[24]. Interestingly, consistent rightward shifts of $d_{ij}$ values were observed in response to a subset of compounds (Supplementary Fig. S16), which was similar to the pattern observed in the subset of IBD samples.

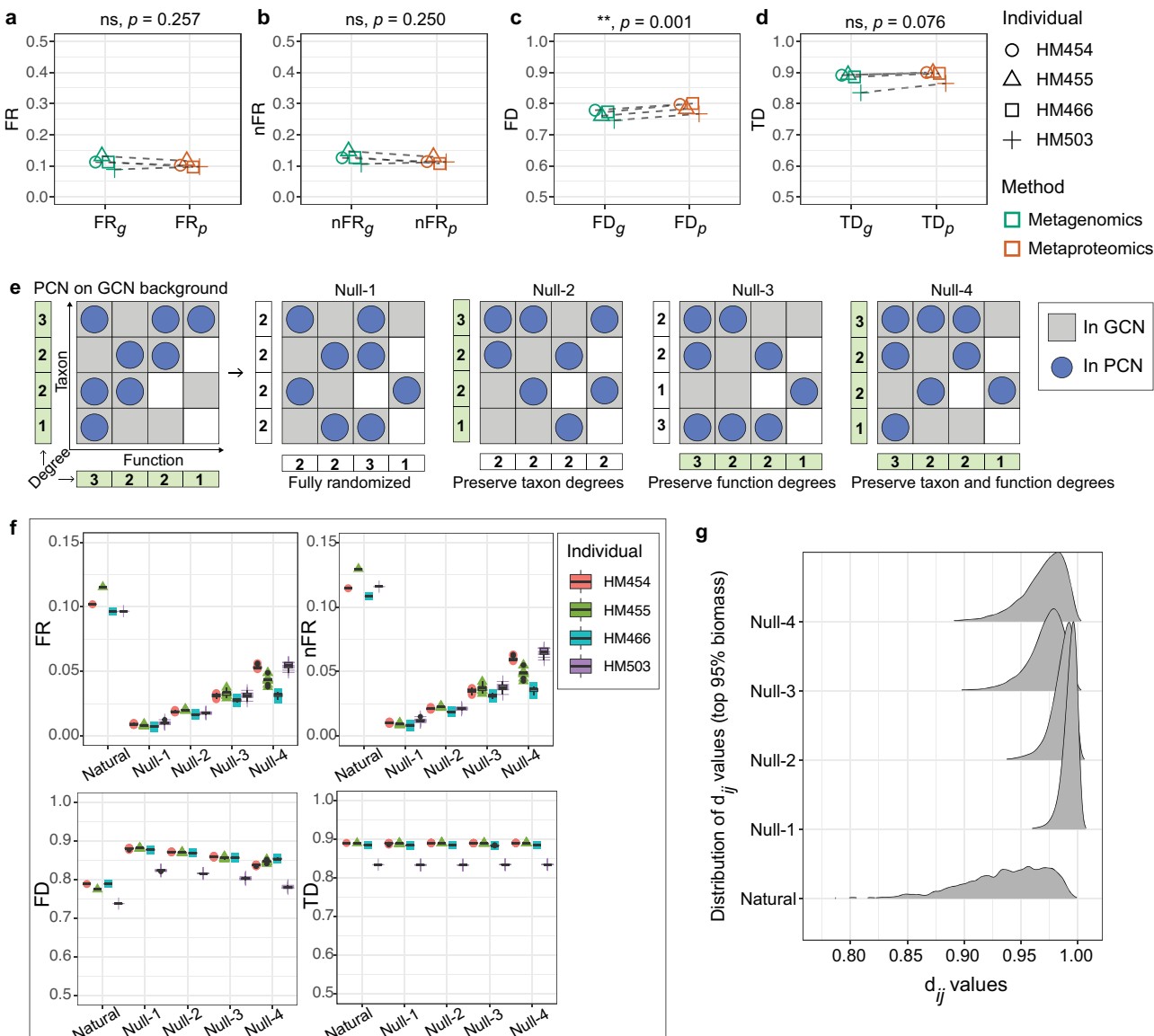

**Fig. 4 | Naturally occurring proteomic-content networks contribute to high functional redundancy in human gut microbiomes. a** Within-sample functional redundancy (FR) in the metagenomes versus in the metaproteomes of the individual microbiomes. **b** Within-sample FR normalized by taxonomic diversity (nFR) in the metagenomes versus in the metaproteomes of the individual microbiomes. **c** Functional diversity (FD) in the metagenomes versus in the metaproteomes. **d** Taxonomic diversity (TD) in the metagenomes versus in the metaproteomes. For (**a**–**d**), paired *t*-tests (two-sided) were performed, ** indicates statistical significance at the $p < 0.01$ level. **e** Scheme diagram showing the creation of four null networks by randomizing a naturally occurred PCN under its corresponding GCN background. Null-1 PCN is a completely randomized network under the GCN background, Null-2 PCN preserves the taxon degrees, Null-3 PCN preserves the function degrees, and Null-4 PCN preserves both taxon and function degrees. **f** Comparison of $FR_p$, $nFR_p$, $FD_p$ and $TD_p$ among naturally occurring PCN and four types of null PCNs in the four individuals. Each of the four null networks was randomized 10 times to demonstrate the reproducibility of the results. Lower and upper hinges correspond to the first and third quartiles, thick line in the box corresponds to the median, and whiskers represent the maximum and minimum, excluding outliers. **g** Distribution of $d_{ij}$ values in natural PCNs and Null-1 ~ 4 PCNs. $d_{ij}$ distributions of genera making up the top 95% protein biomass of the dataset are shown. Source data are provided as a Source Data file.

Similar results were observed in the Berberine dataset (Supplementary Fig. S17 and Supplementary Table S10).

## Discussion

A systems-oriented approach to understanding microbial ecosystems can be employed by constructing networks[11,28,29]. Nevertheless, there has been a substantial gap between interpretating networks constructed from metagenomics and understanding microbiomes' active functionalities. Recently, metaproteomics has experienced exponential growth in its identification coverage[23], providing invaluable deep insights into the expressed functions of microbiomes. In this study, we

demonstrated a method to use metaproteomics dataset to gain a system-level understanding of microbiomes' functionality by computing the functional redundancy of the proteomic content networks.

Using an ultra-deep metaproteomics approach, we showed that the human gut microbiome's taxon-function network on the proteome level is highly nested. In a microbiome PCN, the high nestedness of the network implies that specialist taxa tend to play functional roles that are a subset of active functions from generalist taxa[30,31] Such functional network structures have been frequently found in macro-ecosystem networks of mutualistic interactions (food-webs)[32]. We found that the within-sample FR profiles differed markedly between expressed

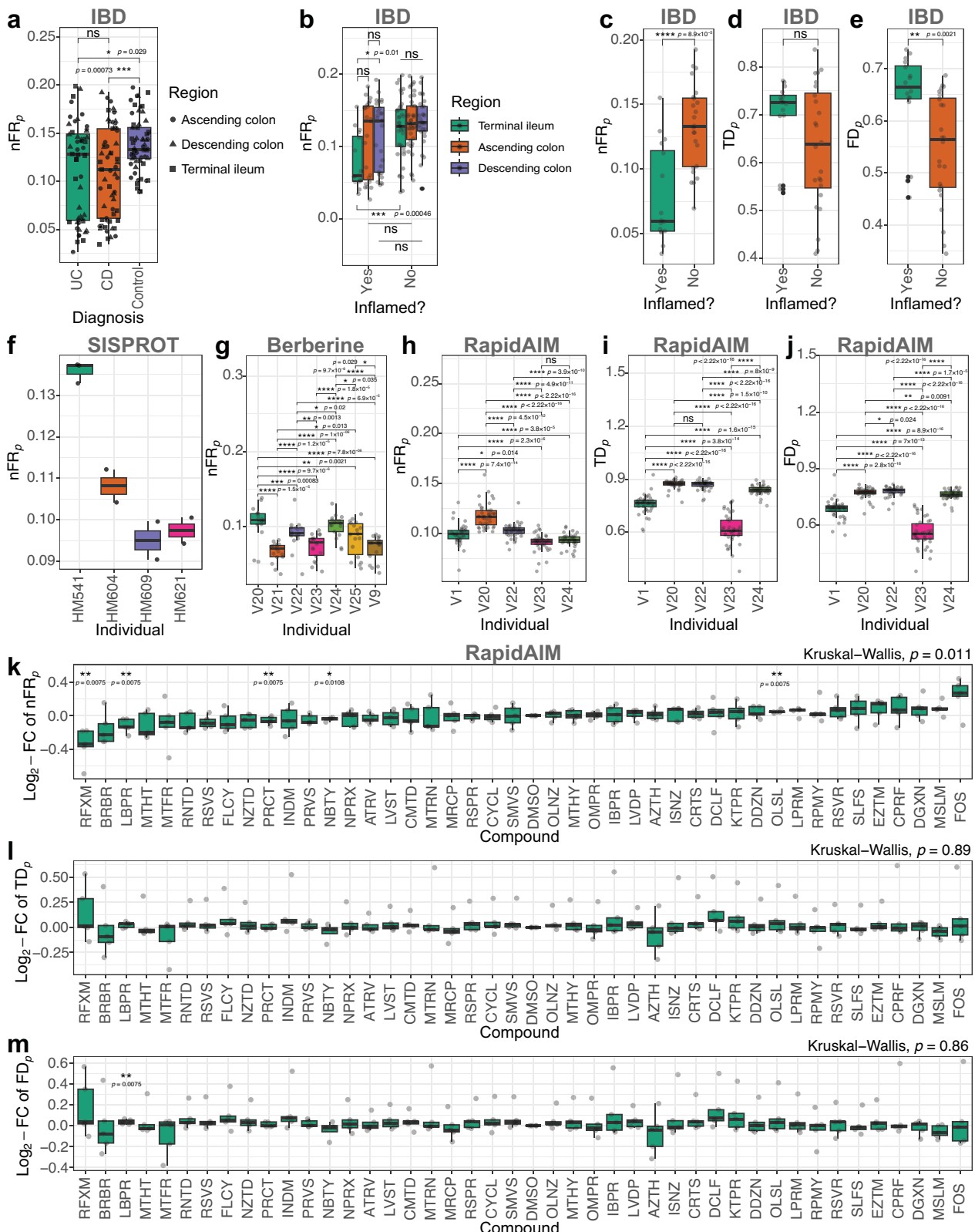

protein functions and genomic functional capacities. Using an ecological model for microbial communities, we demonstrated that such a selective functional expression and the resulting PCN topology favor high richness during community assembly. In the meantime, PCN still maintains high nestedness which contributes to high functional redundancy.

We found that PCNs built by different metaproteomics platforms showed universal properties: datasets generated by shallower analysis approaches still capture the highly nested topology of PCN. This allowed us to make use of routinely generated metaproteomics datasets to observe the effects of multiple environmental factors, such as inter-individual variation, xenobiotics, disease, and biogeography on

**Fig. 5 | Functional redundancy, taxonomic and functional diversity comparisons in different metaproteomics datasets. a** $nFR_p$ values by diagnosis in the IBD dataset, N numbers: $N^{(UC)} = 52$, $N^{(CD)} = 61$, $N^{(Control)} = 63$ samples. **b** $nFR_p$ values by inflammation and gut region in the IBD dataset, N numbers: $N^{(Ascending\ colon,\ inflamed)} = 23$, $N^{(Descending\ colon,\ inflamed)} = 28$, $N^{(Terminal\ ileum,\ inflamed)} = 16$, $N^{(Ascending\ colon,\ non-inflamed)} = 39$, $N^{(Descending\ colon,\ non-inflamed)} = 30$, $N^{(Terminal\ ileum,\ non-inflamed)} = 40$ independent metaproteomic analyses. **c** Comparison of $nFR_p$ values between inflamed and not inflamed regions in TI of CD and control individuals. **d** Comparison of $TD_p$ values between inflamed and not inflamed regions in TI of CD and control individuals. **e** Comparison of $FD_p$ values between inflamed and not inflamed regions in TI of CD and control individuals. N numbers for **C–E**: $N^{(inflamed)} = 16$, $N^{(non-inflamed)} = 24$ independent metaproteomic analyses. **f** $nFR_p$ values by individual microbiomes in the SISPROT dataset. **g** $nFR_p$ values by individual microbiomes in the Berberine dataset, N numbers: $N^{(V20)} = N^{(V21)} = N^{(V24)} = 17$, $N^{(V9)} = N^{(V22)} = N^{(V23)} = N^{(V25)} = 18$ compound treated or control microbiomes. **h** $nFR_p$ values by individual microbiomes in the RapidAIM dataset. **i** $TD_\alpha$ values by individual microbiomes in the RapidAIM dataset. **j** $FD_p$

values by individual microbiomes in the RapidAIM dataset. **k** $Log_2$-fold change (FC) of $nFR_p$ values in comparison to DMSO control samples of each individual (RapidAIM dataset). N numbers, (**h–j**): $N^{(V1)} = N^{(V20)} = N^{(V22)} = N^{(V24)} = 44$, $N^{(V23)} = 43$ compound treated/control microbiomes per individual. **l** $Log_2$-FC of $TD_p$ values in comparison to DMSO control samples of each individual (RapidAIM dataset). **m** $Log_2$-FC of $FD_p$ values in comparison to DMSO control samples of each individual (RapidAIM dataset). N numbers, (**k–m**): N = 5 (with exception $N^{(NBTY)}$ = 4) biologically independent microbiomes. Significance of differences between-groups were examined by Wilcoxon rank-sum test (two-sided); *, **, *** and **** indicate statistical significance at the FDR-adjusted $p < 0.05$, 0.01, 0.001 and 0.0001 levels, respectively. In the box plots, each individual point represents a metaproteomic sample; lower and upper hinges correspond to the first and third quartiles, thick line in the box corresponds to the median, and whiskers represent the maximum and minimum, excluding outliers. In (**k–m**), Kruskall–Wallis test was also performed to detect whether there were significantly variations across all treatment/control groups. Source data are provided as a Source Data file.

the $FR_p$ of the gut microbiome. We first showed that compounds with pharmacological activity can affect the redundancy of expressed functions in individual microbiomes. Overall distributions of functional distances between genera pairs were changed in response to some compounds, which was related to changes in a subset of between-genera functional distances. This suggests that xenobiotic compounds may affect $FR_p$ by partially modifying the functional interrelationship ($d_{ij}$) between proteomes.

Interestingly, in contrast to a partial modification of $d_{ij}$ between proteomes, there was a global shift of $FR_p$ and $d_{ij}$ in a subset of the IBD microbiomes. This finding may support the hypothesis of alternative stable states (bi-stability or multi-stability) in the gut ecosystem[33,34]. One frequently discussed mechanism behind these alternative states has been the continuous exposure of the microbiome to an altered environmental parameter[35]. An inflamed area in the gut will have a reduced mucus layer[36] and elevated host defense responses[26]. The host mucus layer is a nutritional source of cross-feeding in the gut microbiome[37–39]. Loss of this layer decreases the number of available nutrients in the gut environment. We show with our population dynamics model that in agreement with classical ecology[40], the reduction in the number of available environmental nutrients decreases the richness of the microbiome. This could be a factor that contributes to the alteration of functional states. In addition, host defense responses attenuate microbial oxidative stress responses (Pacheco, Osborne and Segrè, 2021), which have been associated with microbiome dysfunction[41]. A decrease in within-sample FR has been associated with impaired microbiome stability and resilience[42]. Resilient microbiota resists external pressures (e.g., antibiotics/dietary shifts) and returns to their original state. Being non-resilient, a microbiome is likely to shift its composition permanently and stay in an altered new state instead of restoring to its original state of equilibrium[43,44]. Collectively, we disassembled the $FR_p$ into one-to-one comparisons of between-taxa functional activities, and found that a global shift in functional roles of microbes toward a more heterogeneous pattern was a factor driving the decrease of $FR_p$ and alteration of states in inflamed areas in IBD patients.

A current limitation of our work is the use of genus-level proteomes. This is because many identified protein groups are shared between different species due to insufficient coverage of species-unique peptide sequences. Future development in the increase of sequence coverage and bioinformatics are warranted. Techniques such as matrix/tensor decomposition[45] or machine learning[46] may support species-level proteome-resolved metaproteomics for a deeper insight into microbiome $FR_p$. Nevertheless, based on the current resolution, we argue that our approach still provides invaluable information on how selective functional expression among taxa shapes the redundancy of expressed functions in a human gut microbiome, and how this redundancy can be affected by the

environment. Future work can focus on revealing the relationship between $FR_p$ and the resilience of a microbiome, so as to gain deeper mechanistic insight for the development of ecological recovery strategies for human gut microbiomes to combat diseases.

## Methods
### Protein extraction and digestion
The sample collection was approved by the Research Ethics Board of the Children's Hospital of Eastern Ontario (CHEO), Ottawa, ON, Canada. Written informed consent form was obtained from their parents. Aspirates of the proximal ascending colon were obtained by colonoscopy, with more details described by Zhang et al.[16]. Protein extraction and digestion of the individual gut aspirate samples were performed as described previously[47], with minor modifications. Frozen aliquots of aspirate samples HM454, HM455, HM466 and HM503 were thawed and subjected to differential centrifugation for microbial cell purification: the samples were first centrifuged at 300 $g$, 4 °C for 5 min to remove debris; the resulting supernatant was centrifuged at 14,000 $g$ for 20 min to pellet microbial cells; the pellets were then washed three times by resuspending in cold phosphate-buffered saline (PBS) and centrifuging at 14,000 $g$, 4 °C for 20 min. Next, the washed microbial cell pellets were resuspended in a cell lysis buffer containing 4% sodium dodecyl sulfate (w/v), 8 M urea, 50 mM Tris-HCl (pH = 8.0), and one Roche cOmplete™ mini tablet per 10 mL buffer, followed by ultra-sonication (30 s on, 1 min off, amplitude of 25%, two rounds) using a Q125 Sonicator (Qsonica, LLC). Cell debris was then removed by a centrifugation at 16,000 $g$, 4 °C for 10 min.

Each of the protein extract was then precipitated in five times its volume of precipitation solution (acetone: ethanol: acetic acid = 49.5: 49.5: 1, v-v:v) at −20 °C overnight. The precipitated proteins were pelleted by centrifuging at 16,000 $g$, 4 °C for 20 min, followed by being washed with ice-cold acetone for three times to remove excess SDS that may affect trypsin activity. Next, the washed proteins were resuspended in a buffer containing 6 M urea and 1 M Tris-HCl (pH = 8.0). Protein concentration was determined by the DC™ assay (Bio-Rad Laboratories, Canada) following the manufacturer's manual.

Finally, proteins were subjected to an in-solution tryptic digestion. The samples were reduced in 10 mM dithiothreitol (DTT) at 56 °C for 30 min, then were alkylated by 20 mM iodoacetamide (IAA) at room temperature in dark for 40 min. The samples were then diluted 10 times with 1 M Tris-HCl buffer (pH = 8.0), followed by trypsin digestion (at a concentration of 1 µg trypsin per 50 µg proteins) at 37 °C for 24 h. The digests were then acidified to pH = 3 using 10% formic acid, followed by a desalting step using Sep-Pak C18 Cartridge (Waters, Milford, MA, USA). The cartridges were first activated using 100% acetonitrile, and then equilibrated using 0.1% formic acid (v/v) before passing samples through the columns for three times. Samples bonded to the cartridges were then washed using 0.1% formic acid (v/v), and

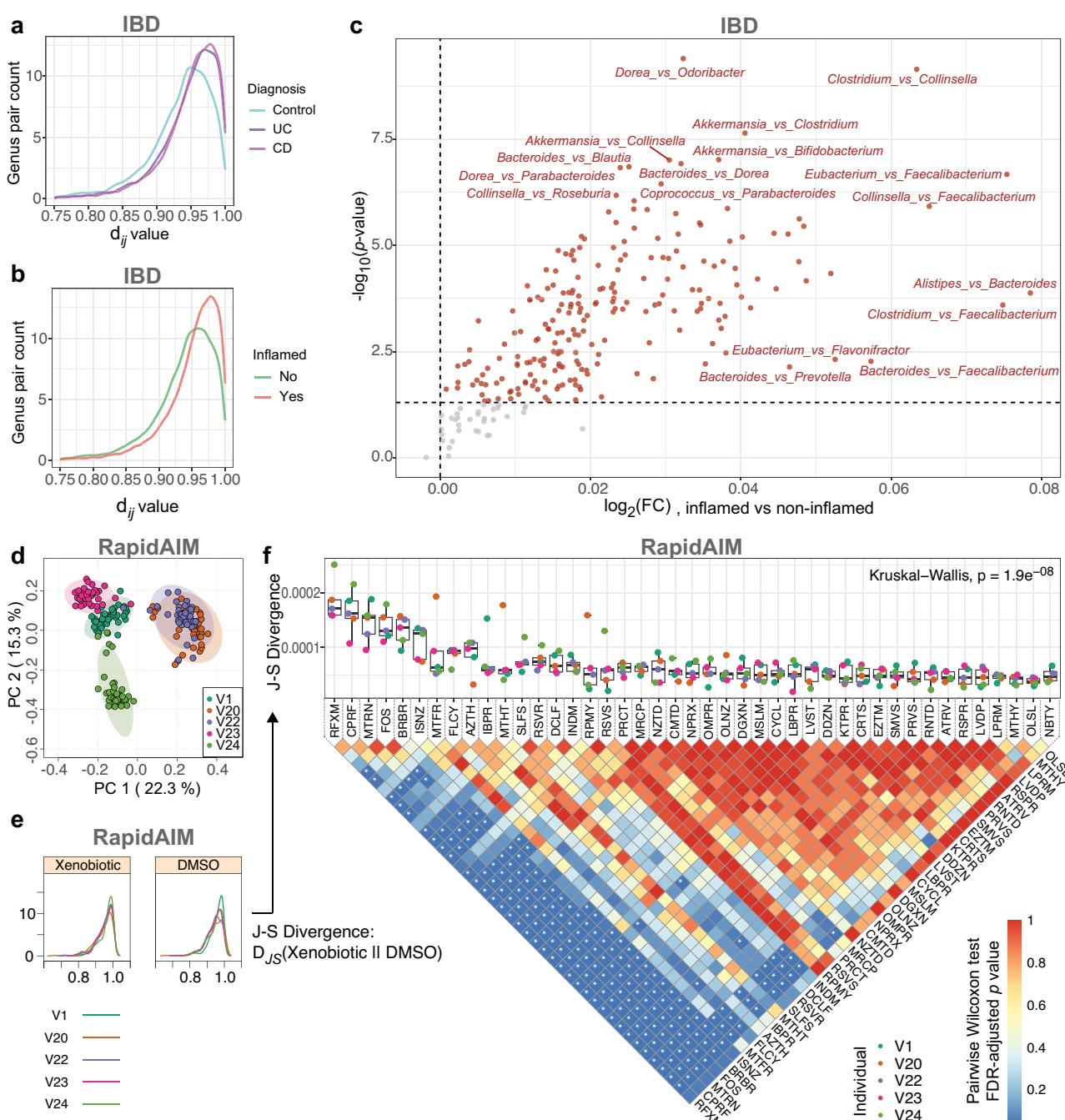

**Fig. 6 | Between-proteome functional distances in IBD and drug-treated microbiomes. a** Distribution of $d_{ij}$ values by diagnosis. **b** Distribution of $d_{ij}$ values by inflammation. **c** Volcano plot showing altered $d_{ij}$ values between inflamed and non-inflamed sampling sites. The results were based on microbial genera of the top 95% overall protein biomass in the dataset. Wilcoxon rank-sum test (two-sided) was performed and $p < 0.05$ was selected as the threshold. **d** Principal component analysis based on between-genera functional distances in individual metaproteomes. **e** Scheme diagram of comparing the dij distribution between drug-treated microbiome and the DMSO control. **f** J-S divergence between the $d_{ij}$ distribution in the control (DMSO) and that of the other compounds (lower and upper hinges

correspond to the first and third quartiles, thick line in the box corresponds to the median, and whiskers represent the maximum and minimum, excluding outliers), $N = 5$ (with exception $N^{(NBTY)} = 4$) biologically independent microbiomes. Kruskal–Wallis test result indicates that overall the compounds had heterogeneous levels of J-S divergence with the DMSO. Between-compound comparisons of the J-S divergence values were performed by a Pairwise Wilcoxon Rank-Sum Tests, * indicates statistical significance at the FDR-adjusted $p < 0.05$ level. The results were based on microbial genera of the top 95% overall protein biomass in the dataset. Source data are provided as a Source Data file.

finally the samples were eluted from the cartridges using the elution solution containing 80% acetonitrile and 0.1% formic acid (v/v).

### High-pH reversed-phase fractionation
Eluted samples were evaporated in a SAVANT SPD1010 SpeedVac Concentrator (Thermo Fisher Scientific, USA), and resuspended in

0.1% formic acid (v/v) to a concentration of 1 μg/μL for high-pH reversed-phase fractionation following a previous workflow[15], with minor adaptations: 30 μL sample were loaded to a ZORBAX Bonus-RP column (with 3.5 μm C18 resins, ID 2.1 mm, length 50 mm; Agilent Technologies, USA), and fractionated with an Agilent 1200 series HPLC System (Agilent Technologies, Germany). A 60 min gradient consisting

of 5–35% buffer B (v/v) in 1–42 min, and 35–50% buffer B in 42–45 min at a flow rate of 100 µL/min was used for the fractionation. Here, 10 mM ammonium formate was used as buffer A, and 10 mM ammonium formate with 90% acetonitrile (v/v) was used as buffer B. Ammonium hydroxide was used to adjust the pH of both buffers A and B to 10. Sample fractions were continuously collected into 96 well plates by an Agilent 1100 Series Micro-FC G1364D micro fraction collector (Agilent Technologies, Germany). For each sample, 48 fractions were collected into different wells at 1 min intervals over the first 48 min. The samples were then pooled by combining four fractions at an interval of 12 wells, resulting in 12 fractionated samples per individual microbiome (Fig. 1a).

## HPLC-ESI-MS/MS analysis

After evaporation and resuspension in 0.1% formic acid, each fraction was analyzed by HPLC-ESI-MS/MS consisting of an UltiMate 3000 RSLCnano system (Thermo Fisher Scientific, USA) and an Orbitrap Exploris 480 mass spectrometer (Thermo Fisher Scientific, USA). A 60 min gradient of 5– 35% (v/v) buffer B at a 300 µL/min flow rate was used to separate the peptides on a tip column (75 µm inner diameter × 10 cm) packed with reverse phase beads (3 µm/120 Å ReproSil-Pur C18 resin, Dr. Maisch GmbH, Ammerbuch, Germany). Here, 0.1% formic acid (v/v) was used as buffer A, and 0.1% formic acid with 80% acetonitrile (v/v) was used as buffer B. The MS full scan ranging from 350 to 1400 m/z was recorded in profile mode with the resolution of 60,000. Data-dependent MS/MS scan was performed with the 12 most intense ions with the resolution of 15,000. Dynamic exclusion was enabled for a duration of 30 s with a repeat count of one. Raw data was collected using Thermo Fisher Scientific Xcalibur™ Software (version 4.4).

## Database search

Database search for the fractionated metaproteomics samples was performed based on the MetaPro-IQ workflow[16]. Briefly, a two-step database search was first performed using X!Tandem (version 2015.12.15.2). All sample fractions were searched against the IGC database of human gut microbiome (http://meta.genomics.cn/)[17] to generate a reduced database, then a classical target-decoy database search was performed using the reduced database to generate confidently identified peptide and protein lists based on a strict filtering criteria of FDR = 0.01. The protein lists for all sample fractions were then combined, and duplicated proteins were removed to generate a combined non-redundant FASTA database using an in-house PERL script. Next, MaxQuant (version 1.5.2.8) was used to generate quantified protein groups and peptides in each sample using the combined non-redundant FASTA database. Carbamidomethylation of cysteine was set as a fixed modification, oxidation of methionine and N-terminal acetylation were set as potential modifications. The maximum missed cleavages of trypsin was set as two. The resulting peptide and protein group lists from MaxQuant were then inputted to MetaLab (version 1.2.0) for taxonomic analysis and functional annotation[48]. For the taxonomic analysis, identified peptides were mapped to taxonomic lineages based on a built-in pep2tax database in MetaLab. Functional annotation to COG[49] was performed using Diamond (version0.8.35). In addition, KEGG KOs were annotated using GhostKOALA[50]. The dataset was deposited to the ProteomeXchange Consortium (http://www.proteomexchange.org) via the PRIDE partner repository with the dataset identifier PXD027297.

For the metagenomics analysis, matched metagenome data of samples HM454, HM455, HM466 and HM503 were obtained from the previous MetaPro-IQ study[16], accessible from the NCBI sequence read archive (SRA) under the accession of SRP068619. To enable the comparison between GCN and PCN, we used the metagenomics data to generate a protein database for another database search. Briefly, the raw reads were processed using MOCAT for trimming and quality

filtering, and for human reads removal as previously described[16]. Next, reads were assembled by MEGAHIT (https://github.com/voutcn/megahit) into contigs. Gene prediction from the contigs were performed using Prodigal (https://github.com/hyattpd/Prodigal) to generate FASTA files. Combined FASTA files were then used for metaproteomic database search following the MetaPro-IQ pipeline.

## Metaproteomic and metagenomic content networks

For the generation of proteomic content networks (PCNs), we developed a 'peptide-protein bridge' approach for the generation of IGC-based PCNs (see details in Supplementary Notes 1).

For the 'peptide-protein bridge' approach, we matched functions to taxa based on four database search output files, i.e., peptides, protein groups, taxonomy and function. The protein groups table (generated by MaxQuant) contains information on the identified proteins, and identifiers of peptide sequence associated to each protein group. The taxonomy table generated by MetaLab contains peptide sequences and their corresponding lowest common ancestor (LCA) taxa. The function table contains identified proteins and their corresponding functional annotations. Proteins were first matched to KOs, then for those that missed KO annotations, COG information were supplemented. Therefore, at first, we matched the protein groups to taxa through the peptides. Next, functions of the proteins were combined to the list to generate a taxon-to-function table that was bridged by the peptide-protein identification relationship. Protein group intensity was used as the quantification information in PCNs.

To generate GCNs from the metagenomics result, the same functional annotation method was used to annotate identified proteins to KEGG-COG. The taxonomic information of proteins was obtained by blasting the Prodigal-predicted sequences against the UHGP V2.0 database[51]. The count of raw reads corresponding to each protein was used as the quantification information in GCNs. Metagenomics-database search based PCNs were generated accordingly using the taxonomic and functional annotation strategy as the GCN.

Next, a PCN of $S$ taxa and $F$ functions can then be represented by an $S \times F$ incidence matrix $\mathbf{P} = [P_{ia}]$, where $P_{ia} \geq 0$ is the total intensity of proteins of function-$a$ in taxon-$i$ normalized by the total intensities of functional proteins in taxon-$i$. Similarly, the GCN can then be represented as $\mathbf{G} = [G_{ia}]$, where $G_{ia} \geq 0$ is the raw read counts of proteins of function-$a$ in taxon-$i$ normalized by the total counts of raw reads in taxon-$i$.

## Calculation of functional distance and functional redundancy

Weighted Jaccard distance $d_{ij}$ between metagenomic (or metaproteomic) contents of taxon-$i$ and $j$ can then be calculated with the GCN and PCN profiles $\mathbf{G}$ and $\mathbf{P}$, respectively, as described previously (Tian et al.[11],). For GCN, we have

$$d_{ij} = 1 - \frac{\sum_a \min(G_{ia}, G_{ja})}{\sum_a \max(G_{ia}, G_{ja})}, \tag{3}$$

and for PCN, we have

$$d_{ij} = 1 - \frac{\sum_a \min(P_{ia}, P_{ja})}{\sum_a \max(P_{ia}, P_{ja})}. \tag{4}$$

The relative abundance of taxon-$i$ in each community was denoted as $p_i$. In each metagenomics sample, $p_i$ was quantified using MetaPhlAn3 with default settings. In each metaproteomics sample, $p_i$ was quantified using the total abundances of unique peptides corresponding to taxon-$i$.

With the $d_{ij}$ and $p_i$ values, within-sample FR of the metagenomic and the metaproteomic profiles, denoted as $FR_g$ and $FR_p$, respectively, were then calculated according to Eq. (1) given in the Results section.

## Consumer-resource model with cross-feeding interactions

Consider a microbiome with a resource pool of $M$ nutrients and a pool of $S$ taxa to start the community assembly, each taxon-$i$ consumes resource-$\alpha$ with a consumption flux of $J_{i\alpha}^{\text{in}} = C_{i\alpha}R_\alpha$ (Fig. 2b), where $C_{i\alpha}$ is the resource consumption rate of taxon-$i$ on resource-$\alpha$ (Fig. 2c), and $R_\alpha$ represents the concentration of resource-$\alpha$. Internal metabolic process of taxon-$i$ converts resource-$\alpha$ into two fractions, i.e., a fraction $l$ of consumed resources for byproduct generation, and the remaining fraction $(1-l)$ contributes to the biomass increase of taxon-$i$ (Fig. 2b). Once resource-$\alpha$ is consumed, multiple byproducts (which are also resources) are generated and this is encoded in the byproduct generation matrix $P$, which describes whether resource-$\alpha$ can be converted to a byproduct or resource-$\beta$ (Fig. 2d). To conserve the total amount of resources, we assumed that $\sum_\beta P_{\alpha\beta} = 1$. Overall, the production flux of resource-$\alpha$ by taxon-$i$ is $J_{i\alpha}^{\text{out}} = l\sum_\beta P_{\alpha\beta}J_{i\beta}^{\text{in}}$ which sums over all byproduct generation activities from all consumed resources. Resource-$\alpha$ is externally supplied with a supply rate $h_\alpha$. All resources and microbes are diluted by the same dilution rate $D$. Overall, the dynamics of the resource concentrations ($R_\alpha$, $\alpha = 1, \ldots, M$) and taxa abundances ($N_i$, $i = 1, \ldots, S$) are given by a set of coupled ordinary differential equations:

$$\frac{dR_\alpha}{dt} = h_\alpha - DR_\alpha + \sum_i N_i(J_{i\alpha}^{\text{out}} - J_{i\alpha}^{\text{in}}), \tag{5}$$

$$\frac{dN_i}{dt} = N_i\left[Y(1-l)\sum_\alpha J_{i\alpha}^{\text{in}} - D\right]. \tag{6}$$

$Y$ is the resource yield. For simplicity, we assume $Y$ is the same for all resources across taxa. Note that the model in Marsland et al.[20] considers two separate dilution rates for resources and microbes. Here, under the assumption of the fed-batch culture, dilution rate $D$ is the same for both resources and microbes.

Simulations of the Consumer-Resource Model with cross-feeding interactions were performed by considering dynamics of 100 microbial taxa and 100 resources. To simulate functionalities represented by the GCN and PCN, the consumption matrices that derive from the GCN ($C^{GCN}$) or the PCN ($C^{PCN}$) are assumed to have the same connectance as the GCN and the PCN respectively. More specifically, we used a binomial distribution with the probability equal to the connectance of the GCN to determine the number of consumable resources for each species and thus formed the incidence matrix of $C^{GCN}$. The consumption matrix $C^{PCN}$ is generated via a subsampling of the $C^{GCN}$ with the subsampling probability to keep the connectance of $C^{PCN}$ the same as the connectance of the PCN. The consumption rates are drawn from the uniform distribution $\mathcal{U}[0,1]$. A universal byproduct generation from consumed nutrients is assumed and its connectance is assumed to be 50%. The entries of byproduct generation matrix are drawn from the uniform distribution between 0 and 1 and even eventually normalized for each consumed nutrient (i.e., $\sum_\beta P_{\alpha\beta} = 1$). Simulations were performed 100 times for each comparison of one pair in the violin plots. For one pair of simulations linked by a thin gray line in the violin plots, two simulations are performed using the same parameters (such as the same byproduct generation matrix) except for the consumption matrix (i.e., $C^{GCN}$ and $C^{PCN}$). The richness of the assembled communities is counted as the number of microbial taxa that have positive abundances at the end of the simulation.

## Statistical analysis and visualization

The statistical details of analysis can be found in the figure legends and in the main texts, including the statistical tests used and significance criteria. Computation of GCN, PCN and functional redundancy were performed using in-house Python codes. NODF values were computed using the R package RInSp. Jensen-Shannon divergence was calculated

using the R package textmineR. Two-way ANOVA was performed using R function aov(). PERMANOVA tests were performed using R packages "vegan" and "BiodiversityR". Kruskal–Wallis and Wilcoxon rank-sum tests were performed using R functions kruskal.test() and wilcox.test(), respectively. Network incidence matrices, degree distributions, bar plots, box plots, and violin plots were visualized using the R package ggplot2. Unipartite networks were visualized using the R package igraph. Tripartite networks were visualized using the R package networkD3. Heatmaps were visualized using the R package pheatmap. Volcano plot was analyzed by MetaboAnalyst (version 4.0) under non-parametric test setting. The interactive webpage (https://shiny2.imetalab.ca/shiny/rstudio/PCN_visualizer/) for visualization PCNs of metaproteomic datasets analyzed in this paper was created using the R packages shiny and shinydashboard. The Consumer-Resource Model model was implemented in Python 3.8 and Python packages Pandas, Numpy, Seaborn, and matplotlib.pyplot were used.

## Reporting summary

Further information on research design is available in the Nature Portfolio Reporting Summary linked to this article.

## Data availability

The ultra-deep metaproteomics datasets were deposited to the ProteomeXchange Consortium (http://www.proteomexchange.org) via the PRIDE partner repository with the dataset identifier PXD027297. Database search outputs from the SISPROT[23], RapidAIM[24], Berberine[25] and IBD[26] studies have been previously deposited to the ProteomeXchange Consortium with the dataset identifiers PXD005619, PXD012724, PXD015934 and PXD007819, respectively. Proteomics dataset of the cultured singles strain samples (Wang et al.[19],) has been previously deposited to the ProteomeXchange Consortium with the dataset identifier PXD037923. The four metagenomic datasets matching the ultra-deep metaproteomics datasets were obtained from the previous MetaPro-IQ study[16], accessible from the NCBI sequence read archive (SRA) under the accession of SRP068619. Source data are provided with this paper.

## Code availability

Custom codes for the construction of PCN and calculation of $FR_p$ are available at GitHub: https://github.com/yvonnelee1988/Metaproteome_FRp.

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

## Acknowledgements

This work was partially funded by the Government of Canada through Genome Canada and the Ontario Genomics Institute (OGI-114 and OGI-149) and the Ontario Ministry of Economic Development and Innovation (Project 13440). Y.-Y.L. acknowledges grants from National Institutes of Health (R01AI141529, R01HD093761, RF1AG067744, UH3OD023268, U19AI095219 and U01HL089856). The shotgun metagenomic analysis presented here was enabled in part by WestGrid () and Compute Canada (www.computecanada.ca). D.R.M. is partially supported through University of Ottawa Faculty of Medicine Distinguished Clinical Research Chair in Pediatric Inflammatory Bowel Disease. C.M.A.S. was funded by a

stipend from the NSERC CREATE in Technologies for Microbiome Science and Engineering (TECHNOMISE) Program. The authors acknowledge Ruth Singleton (Clinical Research Coordinator) for participant recruitment and data collection.

## Author contributions

Conceptualization, Y.-Y.L, D.F., L.L., and T.W.; Methodology, D.F., Y.-Y.L, L.L., Z.N., and T.W.; Formal Analysis: L.L.; Investigation L.L., T.W., and Z.N.; Resources, D.R.M., A.S., J.B., and J.M.; Data Curation: L.L., J.B., X.Z., Z.N., J.S., and C.M.A.S.; Writing–Original Draft, L.L., Y.-Y.L, and D.F.; Writing–Review & Editing, J.B., J.M., A.S., C.M.A.S., D.R.M., X.Z., T.W., and Z.N.; Visualization: L.L., T.W., and Z.N.; Supervision: D.F. and Y.-Y.L. The authors declare that they follow principles of inclusion & ethics in global research.

## Competing interests

D.F., A.S., and D.R.M. have co-founded MedBiome, a clinical microbiomics company. All other authors declare no potential competing interests.
