## [Peer Review File · Nature Communications]

REVIEWER COMMENTS

Reviewer #1 (Remarks to the Author):

The paper by Li et al. presents an interesting and innovative novel method to measure functional redundancy in microbiome samples based on metaproteomic analysis coupled to network analysis of taxon-specific proteome content. This is an important topic, and could potentially help to ecologically and functionally interpret differences between microbiome samples. The manuscript is well-written and illustrated with beautiful figures, and the method is interesting and novel. Still, I have a few major concerns.

First and most importantly, the authors use the COG database to measure functional redundancy. This database only contains <5,000 protein families, and many of these families comprise huge numbers of distinct biochemical and ecological functions. The fact that even within single bacterial genomes, many of these COGs have many different functionally distinct copies speaks to the fact that this may be problematic to use as a measure of functional redundancy. Taking cytochrome P450s (COG2124) as an example, this family has hundreds of distinct documented biochemical functions (see [https://www.uniprot.org/uniprotkb?facets=reviewed%3Atrue&query=p450 AND \(taxonomy_name%3Abacteria\)](https://www.uniprot.org/uniprotkb?facets=reviewed%3Atrue&query=p450%20AND%20taxonomy_name%3Abacteria)), which are not at all redundant. I believe this is more the rule than the exception, which makes me suspect that the current method will lead to a gross overestimation of functional redundancy. I am also missing a validation to show that (or quantify to what extent) the metric in fact accurately measures true functional redundancy in for example a model system (perhaps a small synthetic microbiome)?

Second, I am also puzzled by the claim of the authors that species-level analyses were not possible, because many proteins are shared between species (line 380-381). Across species boundaries, proteins will hardly ever have the exact same sequence? It would be very interesting and helpful to see how results at different taxonomic levels would compare.

A few more notes follow below:

- Line 48: 'diversity calculators' -> 'taxonomic diversity calculators'?
- Line 166-170: I wonder whether some of this may be biased by (large) differences in (required) protein abundance from different functional categories, which may cause some proteins simply to be below the detection limit (especially for rarer microbes).

Reviewer #2 (Remarks to the Author):

The paper entitled “Revealing Proteome-Level Functional Redundancy in the Human Gut Microbiome using Ultra-deep Metaproteomics” by Li and Wang brings an innovative use of metaproteomics to assess functional redundancy in the human gut microbiome. They provide an approach to do so, demonstrate that it is applicable to metaproteomic datasets from various methodologies and show that functional redundancy and subsequent metrics are biologically relevant to detect significant microbiome responses to environmental factors.

It was really interesting to read and I do have few comments or questions for clarification. Those few points are summarized below:

First, and as a very broad comment, I wonder if using large unspecific database do not artificially increase functional redundancy (Also see later comment on database usage and metagenomics)? You have performed a 2-step search but of course, for some closely related species that might not be in your sample, you would still have a hit and those species will end up in your second search and lead to increased functional redundancy. Would such measurements not be more accurate using a more specific database?

In a related comment, in Figure 3C, FDp is higher than FDg for sample HM454. Please correct me if I'm wrong, but my understanding is that it should be (in theory) impossible to have a FDp higher than a FDg. In practice, it might be possible if things are missed at the metagenomic level or if there are false positives at the metaproteomics level.

Your GCN and PCN are both calculated using matches against the IGC database right? How would you explain that situation?

Since you have metagenomics, why didn't you use it as a database for your Metaproteomics? I guess with that approach you would have not found a FDp higher than FDg.

In the section, Protein-level FR outcompetes diversity indices in detecting microbiome responses to environmental factors:

In lines 256-258, you mentioned that “In patients diagnosed with (IBD), nFRp levels were significantly lower than that of the non-IBD individuals”. What was the cause, was it due to a loss of taxa or a loss of function?

In the section, Alteration of between-proteome functional distances in diseases and compound-treated microbiome:

Dij is the functional distance between taxa i and taxa j (considering their proteome content).

In line 303, you mention that Dij values increased in inflammation. To me, it makes sense that if the distance increases, the redundancy decreases. Yet, in line 304, you write that there is a significant contribution of between proteome functional distance decrease to the overall decrease of nFRp in IBD samples. Either the “proteome functional distance” in that sentence does not correspond to Dij and then I’m a bit confused, or the “proteome functional distance” should increase and the sentence is wrong. Can you clarify please?

Methodology section:

In line 609, you mentioned that Jensen-Shannon divergence and Kullback–Leibler divergence were calculated using the R package LaplacesDemon. However, in the results only KL was used. I actually had initially a random, out of curiosity question: What features of the Dij distributions made you choose KL where people usually go for Jensen-Shannon? Were the results the same when using Jensen-Shannon?

Supplementary figure section:

In supplementary figure 3, you mention the UHGP database. I do not find it in the main text/in the methodology. Am I missing it?

In any case, in supplementary figure 3, the numbers of genera found by either UHGP or IGC are very different, but they are not for COG. Do you have an idea why? And (going back to my first question) what is the effect of the database choice in the assessment of functional redundancy. Do your findings hold if you were to use the UHGP catalogue?

Minor comments/typos:

Line 248: SISPRORT  SISPROT

Line 328: Functionals  functions

Line 334: tend to be playing  tend to play

Line 514-515: searching against the IGC or database What does that mean: OR database?

Figure 2I- Legend: There are not 4 rows. Maybe upper left panel, lower right panel,etc,.. would be clearer.

Responses to Reviewer #1

Point 1.0: The paper by Li et al. presents an interesting and innovative novel method to measure functional redundancy in microbiome samples based on metaproteomic analysis coupled to network analysis of taxon-specific proteome content. This is an important topic, and could potentially help to ecologically and functionally interpret differences between microbiome samples. The manuscript is well-written and illustrated with beautiful figures, and the method is interesting and novel. Still, I have a few major concerns.

Response: We thank Reviewer #1 for reviewing our manuscript and her/his very positive assessment on the general interest and novelty of our method. Next, we address each of the reviewer comments in order. (Figures associated with the responses are shown in the end of this response letter as **Figs. R1-17.**)

Point 1.1: First and most importantly, the authors use the COG database to measure functional redundancy. This database only contains <5,000 protein families, and many of these families comprise huge numbers of distinct biochemical and ecological functions. The fact that even within single bacterial genomes, many of these COGs have many different functionally distinct copies speaks to the fact that this may be problematic to use as a measure of functional redundancy. Taking cytochrome P450s (COG2124) as an example, this family has hundreds of distinct documented biochemical functions (see [https://www.uniprot.org/uniprotkb?facets=reviewed%3Atrue&query=p450 AND \(taxonomy_name%3Abacteria\)](https://www.uniprot.org/uniprotkb?facets=reviewed%3Atrue&query=p450%20AND%20taxonomy_name%3Abacteria)), which are not at all redundant.

Response: We fully agree with Reviewer #1 that some COGs have many distinctly documented functions. In a previous study (Tian et al. (2020)), instead of using the COG database, the authors used the KO (KEGG Orthology) database to calculate the gene-level functional redundancy. In our dataset, while 75.9% of COGs correspond to unique KOs, the remaining 24.1% of COGs were matched to multiple KOs (see **Fig.R1**).

There are two reasons why we chose to use the COG database instead of the KO database in the previous version of our manuscript:

1. **A trade-off between functional granularity and annotation coverage.** Although COG compromises functional granularity, the COG database provides a higher annotation coverage than the KEGG (for example, for the deep metaproteomics dataset of the four individuals, there were a total of 50,216 protein groups identified. 46,095 (91.7%) of these protein groups were successfully annotated with COGs, while only 37,795 (75.3%) of these protein groups were annotated with KOs). Although some annotated functions in COG belong to the categories of 'General function prediction only' and 'Functions unknown', they helped us increase the annotation coverage and thus can be included in our FR_p computation. Therefore, there was a trade-off between using a functional annotation of a **higher coverage** and using a functional annotation of a **higher resolution**.

2. **Lower coverage of functional annotation decreased sensitivity to detect FR_p responses.**

To determine whether **higher coverage** or **higher resolution** is more advantageous, we compared COG vs KEGG in capturing functional distance between different drug treatments, along with CAZyme annotations which also had low functional coverage but high resolution. We found that COG functional annotation is more sensitive than KEGG in detecting responses in microbiome functional networks (more significant differences for the case of COG annotation in **Fig. R2A-C**). Therefore, we chose the COG annotation in the FR_p analysis in the previous version of our paper.

Motivated by Reviewer #1's comment, we took a deeper step into this question and came up with a better solution so that **we do not have to compromise either coverage or resolution. By combining the merit of both KEGG and COG annotations, we can gain a high annotation coverage meanwhile maximizing functional granularity.** Specifically, we first used KEGG to annotate functions to protein groups. Next, for those that could not be annotated with a KO, the annotations were complimented with COG when a protein-COG match was present. In this way, we achieved a high coverage of annotated proteins (92.7%) without compromising the granularity of functions. We thereafter refer to this method as the **KEGG-COG annotation**.

In the revised manuscript, we replaced the COG functional annotation with the KEGG-COG annotation and re-rerun all the data analyses. We found that the previous findings based on COG annotation were well-reproduced by the KEGG-COG annotation (**Fig. R2D; Fig. R2E-F; Figs. 2-5; Supplementary Figs. S10-17; Supplementary Tables S5-10**). For example, as show in **Fig. R2E-F**, previous and revised findings in the IBD dataset are consistent.

To clarify the trade-off between coverage and resolution and introduce the KEGG-COG annotation method, in the revised manuscript, we have added the following sentences (see main text, page 5, lines 127-146):

“In terms of functional annotations, the Kyoto Encyclopedia of Genes and Genomes (KEGG) database has been widely used in functional metrics such as the genomic-level functional redundancy (Tian *et al.*, 2020). However, it is common in metaproteomic studies that a certain proportion of proteins does not have a KEGG annotation. Indeed, in this dataset, there were a total of 50,216 protein groups identified, among which, 46,095 (91.7%) were successfully annotated with clusters of orthologous groups (COGs), while only 37,795 (75.3%) were annotated with KEGG Orthologs (KOs). Therefore, we complemented the KEGG annotations with COG to achieve a better coverage (denoted as KEGG-COG annotation; see **Supplementary Note S2** for more comparisons). This annotation will be applicable to metaproteomic-based functional redundancy computations without the need for the samples' metagenomes.

In addition, to facilitate direct comparisons of redundancy or network metrics between the GCN and the PCN, we further used the samples' paired metagenomes to generate prodigal-predicted protein sequences as the database to perform another metaproteomic database search. An average of 65,541 unique peptides and 29,392 protein groups per sample were

obtained from the search. The prodigal sequences were blasted against the UHGP database for taxonomic matches. KEGG-COG annotations were performed. GCNs or PCNs were then computed by summing read counts or protein intensities at each taxon-function incidence (see **Methods**).”

Point 1.2: I believe this is more the rule than the exception, which makes me suspect that the current method will lead to a gross overestimation of functional redundancy.

Response: We thank Reviewer #1 for this critical comment. We compared the nFR_p values calculated from COG, KEGG, and KEGG-COG annotations (**Fig. R3**). As Reviewer #1 predicted, the COG showed a slight overestimation of nFR_p . However, despite better functional granularity, KEGG annotation alone results in even higher nFR_p . Therefore, the lower coverage of proteins using KEGG may induce an even high impact on nFR_p calculation. We found that the new KEGG-COG annotation decreased nFR_p by solving the resolution and coverage issues at once. We further validated the slight overall decrease of nFR_p for the KEGG-COG annotation using the IBD dataset (**Fig. R4**). Note that the overall difference did not affect any conclusion on the comparison between groups in our dataset (e.g., **Fig. R2**).

Point 1.3: I am also missing a validation to show that (or quantify to what extent) the metric in fact accurately measures true functional redundancy in for example a model system (perhaps a small synthetic microbiome)?

Response: We thank Reviewer #1 for this excellent suggestion. We further demonstrated the sensitivity and strength of FR_p and nFR_p using *in silico* communities generated with genomes and proteomes of six single bacterial strains (i.e., *Phocaeicola (Bacteroides) vulgatus* ATCC 8482, *Bacteroides ovatus* ATCC 8483, *Bacteroides uniformis* ATCC 8492, *Blautia hydrogenotrophica* DSM 10507, *Escherichia coli* DSM 101114). Proteomes of these strains cultured in four media (basal medium with or without added sugars glucose, sucrose, or kestose) were obtained from our previous study (Wang et al., 2022, doi: <https://doi.org/10.1101/2022.11.04.515228>) (**Fig. R5A**). We first used the genomes and proteomes (in basal media) to generate different three-member communities *in silico* (**Fig. R5B, C**). When all three members belong to the *Bacteroides* genus (or *Phocaeicola* genus), both genome- and proteome-level functional redundancy were higher compared with the other combinations. The redundancies decreased as the community becomes more diverse at the genus level. The agreement of changing trends between genome- and proteome-level functional redundancy (**Fig. R5B vs. R5C**) to some extent validates the accuracy of FR_p and nFR_p .

Furthermore, we argue that despite genome-level functional redundancy FR_g seems predictive of the proteome-level functional redundancy FR_p here, in principle, FR_g only responds to the change of microbial abundances. The responses of proteomes to the environment cannot be fully captured by genomes. However, accurate proteomes cannot be easily derived *in-silico*. To

demonstrate the effect of variable proteomes on FR_p and nFR_p , we replaced the proteomes of strains with the ones cultured in the presence of different sugars (and maintained microbial abundances), finding that the levels of FR_p and nFR_p showed fluctuations. Note that since the microbial compositions were the same among groups, their genome-level functional redundancies were equal (**Fig. R5D**). This suggests that FR_p and nFR_p are sensitive to a community's functional responses, even if induced solely by proteome changes. This further clearly emphasizes the value of performing proteome-level functional redundancy analysis as an important community ecology metric beyond metagenome-based approaches.

This result has been added to the revised manuscript as **Supplementary Figure S3** and the corresponding texts have been added (page 7, lines 178-197):

“We demonstrate the sensitivity of FR_p and nFR_p using *in silico* communities generated with genomes and proteomes of single bacterial strains (i.e., *Phocaeicola (Bacteroides) vulgatus* ATCC 8482, *Bacteroides ovatus* ATCC 8483, *Bacteroides uniformis* ATCC 8492, *Blautia hydrogenotrophica* DSM 10507, *Escherichia coli* DSM 101114). Proteomes of these strains cultured in four different media (basal medium with or without added sugars glucose, sucrose or kestose) were obtained from our previous study (Wang *et al.*, 2022) (**Supplementary Figure S3A**). We first used the genomes and proteomes (in basal media) to generate different three-member communities *in silico* (**Supplementary Figures S3B and S3C**). When all three members belong to *Bacteroides* or *Phocaeicola* genera, the community's genome- and proteome-level functional redundancy were both higher compared with the other combinations. The redundancies decreased as the community becomes more diverse at the genus level. In **Supplementary Figures S3B and S3C**, despite genome-level functional redundancy may seem predictive of the proteome-level functional redundancy, we emphasize that, in principle, genome-level functional redundancy only responds to the change of microbial abundances. When we further replaced the proteomes of strains with the ones cultured in the presence of different sugars (and maintained microbial abundances), the levels of FR_p and nFR_p showed fluctuations (**Supplementary Figure S3D**). This suggests that FR_p and nFR_p are sensitive to a community's functional responses, even induced solely by proteome alterations while microbial abundances are unchanged.”

Point 1.4: Second, I am also puzzled by the claim of the authors that species-level analyses were not possible, because many proteins are shared between species (line 380-381). Across species boundaries, proteins will hardly ever have the exact same sequence?

Response: We apologize for not explaining it more clearly and explicitly. It is true that proteins from different species will hardly have the exact same sequence. The reason why LC-MS/MS identified protein(Group)s are shared between species is that the metaproteomics approach identifies proteins relying on tryptic peptides. As shown in **Fig. R6**, while LC-MS/MS identified the

peptide sequences (highlighted in cyan, green, and yellow), it missed the sequences (highlighted in red) that distinguish the two species from the same genus (taken from the four individuals' metaproteomics dataset). Therefore, the current limitation in the metaproteomics techniques renders the species-level FR_p computation infeasible. We hope future advancement in sequence coverage and bioinformatics will facilitate the species-level FR_p calculation. We have mentioned this point in the revised manuscript (see Page 15, lines 423-426):

“This is because many identified protein groups are shared between different species due to insufficient coverage of species-unique peptide sequences. Future advancements in sequence coverage and bioinformatics are warranted.”

Point 1.5: It would be very interesting and helpful to see how results at different taxonomic levels would compare.

Response: We thank Reviewer #1 for this excellent suggestion. Here, we selected the rifaximin (RFXM) treatment vs DMSO groups in the RapidAIM dataset to compare results at different taxonomic levels. From the genus level to the phylum level, a gradual increase in nFR_p was observed (**Fig. R7**). Although the RFXM group showed an overall decline in nFR_p compared to the DMSO group, a significant decrease (one-sided Wilcoxon test, $p < 0.05$) was only observed at the genus level. This suggests that the genus-level nFR_p provides a good sensitivity in detecting the community response to environment changes.

Point 1.6: A few more notes follow below:

- Line 48: 'diversity calculators' -> 'taxonomic diversity calculators'?

Response: We have corrected it to 'taxonomic diversity calculators'.

Point 1.7: - Line 166-170: I wonder whether some of this may be biased by (large) differences in (required) protein abundance from different functional categories, which may cause some proteins simply to be below the detection limit (especially for rarer microbes).

Response: We thank Reviewer #1 for this comment. It is possible that proteins below the detection limit can be missed in metaproteomics analysis. However, in Fig. 1A, the statement was based on a summary of proteins on the phylum-COG category level, meaning that RNA processing and modification (A) and mobilome (X) proteins were indeed overall too low to be identified from the dataset, suggesting an overall low abundance of their presence in the community.

Finally, we thank Reviewer #1 again for reviewing our manuscript and her/his very insightful comments that help us significantly improve the quality of our work. We hope our responses above have addressed her/his comments in a satisfactory manner.

Response to Reviewer #2

Point 2.0: The paper entitled “Revealing Proteome-Level Functional Redundancy in the Human Gut Microbiome using Ultra-deep Metaproteomics” by Li and Wang brings an innovative use of metaproteomics to assess functional redundancy in the human gut microbiome. They provide an approach to do so, demonstrate that it is applicable to metaproteomic datasets from various methodologies and show that functional redundancy and subsequent metrics are biologically relevant to detect significant microbiome responses to environmental factors. It was really interesting to read and I do have few comments or questions for clarification. Those few points are summarized below:

Response: We truly appreciate Reviewer #2’s positive assessments of our work. We have provided point-by-point responses to Reviewer #2’s comments below, with important figures being attached at the end of the response letter as **Figs. R1-17**.

Point 2.1: First, and as a very broad comment, I wonder if using large unspecific database do not artificially increase functional redundancy (Also see later comment on database usage and metagenomics)? You have performed a 2-step search but of course, for some closely related species that might not be in your sample, you would still have a hit and those species will end up in your second search and lead to increased functional redundancy. Would such measurements not be more accurate using a more specific database?

Response: We thank Reviewer #2 for this very insightful comment. We interpret it as follows: if we use a large unspecific database covering much more species than those in the actual sample, we may detect peptides shared among proteins from different species and therefore get an overestimated list of possible protein IDs in each proteinGroup. Using a 2-step search, we were able to shorten the list of proteins by generating a reduced database from the large unspecific database. Reviewer #2 pointed out that such a database may still include species that are absent in the samples, and therefore proteins from those absent species may still end up in the identified proteinGroups, which may affect the assessment of functional redundancy.

We addressed Reviewer #2’s concern as follows. First, we fully agree with Reviewer #2 that using a dataset-specific metagenomic database can eliminate such a problem. We performed this analysis and did not find a big increase in functional redundancy (**Figs. R8-9**; please also see our detailed responses to **Point 2.3** raised by Reviewer #2). Second, we developed the ‘protein-peptide-bridge’ approach which can avoid the above issue. In the ‘protein-peptide-bridge’ approach, a taxonomic match is not performed through protein IDs in the proteinGroups. As a result, it would not include absent species due to any presence of their protein IDs. Instead, we used a peptide-centric approach to link proteinGroups to taxa. Peptides derived from confident (FDR = 1%) peptide-spectrum matches were matched to their taxonomically lowest common ancestors, and their corresponding proteinGroups were given taxonomic assignments accordingly. In this way, interference from absent species in the database would not exist. The

main advantage of our ‘protein-peptide-bridge’ approach is that it is applicable when no matched metagenomic dataset is available in a metaproteomic study.

Point 2.2: In a related comment, in Figure 3C, FD_p is higher than FD_g for sample HM454. Please correct me if I’m wrong, but my understanding is that it should be (in theory) impossible to have a FD_p higher than a FD_g . In practice, it might be possible if things are missed at the metagenomic level or if there are false positives at the metaproteomics level. Your GCN and PCN are both calculated using matches against the IGC database right? How would you explain that situation?

Response: We thank Reviewer #2 for this comment. It is possible that FD_p is higher than FD_g , at least with our FD metric based on Rao’s quadratic entropy. Here, we use *in silico* communities simulated by using proteomic and genomic data of single strains to demonstrate why FD_p can be higher than FD_g . According to the FD definition: $FD = \frac{\sum_{i=1}^S \sum_{j \neq i}^S d_{ij} p_i p_j}{\sum_{i=1}^S p_i}$, where $d_{ij} = 1 - \frac{\sum_a \min(G_{ia}, G_{ja})}{\sum_a \max(G_{ia}, G_{ja})}$ for GCN, and $d_{ij} = 1 - \frac{\sum_a \min(P_{ia}, P_{ja})}{\sum_a \max(P_{ia}, P_{ja})}$ for PCN, the FD_p and FD_g values are affected by multiple factors, i.e., the relative abundance of each taxon, the presence/absence of each function, and the abundance of each function. In real natural microbiomes, all these factors impact FD values. As shown in **Fig. R5** of this response letter (and in **Supplementary Figure S3** of the revised manuscript), we performed proteomics analysis based on culture results of six single bacterial strains, and *in silico* created different communities using the proteomes and genomes. Our results showed that all *in silico* communities had higher FD_p than FD_g .

To create an even simpler example, we eliminated the contribution of proteomic and genomic functional abundances by replacing the abundance values with 1 (presence) or 0 (absence). Therefore, now we could calculate the unweighted- FD_p which is solely affected by the presence/absence of functions in each taxon. Results still showed higher (unweighted-) FD_p than (unweighted-) FD_g values in the simulated communities (**Fig. R10**).

Point 2.3: Since you have metagenomics, why didn’t you use it as a database for your Metaproteomics? I guess with that approach you would have not found a FD_p higher than FD_g .

Response: We thank Reviewer #2 for this comment. In the original manuscript, we did not use metagenomic data to generate a protein database for the ultra-deep metaproteomic dataset, we used the IGC database instead. From the perspective of providing a practical computational workflow for researchers, we were aware of that in many metaproteomic studies there may not be accompanying metagenomic data available. Therefore, we used the “protein-peptide-bridge” approach to generate PCNs from stand-alone metaproteomic datasets. This approach is therefore applicable to the four other datasets (SISPROT, RapidAIM, Berberine, and IBD) that didn’t have metagenomes sequenced.

We fully agree with Reviewer #2 that using metagenomics-based protein databases will facilitate a better comparison between a microbiome's GCN and PCN. Therefore, in the revised manuscript, we used metagenomics to generate prodigal-predicted protein sequences and used it as the database to search the ultra-deep metaproteomics dataset. We showed that the comparisons of FR, nFR, TD and FD between GCNs and PCNs remained similar. The PCNs still preserved a high level of FR and nFR from their corresponding GCNs, and we still found FD_p values higher than FD_g (**Fig. R8**).

We also successfully reproduced other results based on GCN-PCN networks. These include the connectance of new GCNs and PCNs. Previously, we used GCN connectance = 0.220 and PCN connectance of 0.049 to perform the simulation of the Consumer-Resource Model (CRM) (**Fig. 2B-G**). Using the new search results, we obtained the same values of connectance of GCN (0.22 ± 0.02 ; Mean \pm SD, N = 4) and PCN (0.05 ± 0.02 ; Mean \pm SD, N = 4). Therefore, we do not need to change the parameters and results of the CRM models.

We also randomized the new PCNs to generate the four types of null networks. Our new results are consistent with our previous findings based on the IGC search, where real PCNs had higher levels of FR_p and nFR_p than their randomized counterparts (**Fig. R9A vs Fig. R9C**). The comparisons of d_{ij} distributions in different groups of networks looked also very similar (**Fig. R9B vs Fig. R9D**). Therefore, the metagenome-based results did not affect our findings and conclusions.

In the revised manuscript, we have added the metagenome-based results (**Figure 2H-I, Figure 3, and Supplementary Figures S4, S10**) and added texts (page 7, lines 200-203):

“The PCNs of metagenome and IGC databases-based search yielded similar depth, both achieved reasonable depths compared with each individual's respective GCNs (**Supplementary Figure S4**)”.

In addition, we have modified the texts in the revised manuscript (pages 8-9, lines 231-244):

“By visualizing incidence matrices of these PCNs, we observed highly nested structures (**Figure 2H and Supplementary Figure S10**) and found that the Nestedness metric based on Overlap and Decreasing Fill (NODF) were high in the PCNs ($NODF = 0.28 \pm 0.01$; Mean \pm SD, N = 4, metagenome database-based search), which are close to those of the respective GCNs ($NODF = 0.36 \pm 0.05$; Mean \pm SD, N = 4). Similarly, the PCNs based on IGC database-based search also resulted in high NODF values (0.34 ± 0.01 ; Mean \pm SD, N = 4). We then calculated the degree distributions of genera and functions in the PCNs and GCNs, respectively. On the functional dimension, similar to previous observations in GCNs (Tian et al., 2020), the degree distributions of functions in both the GCN and PCN have fat tails, represented by some functions being associated with a high number of taxa (**Figure 2I and Supplementary Figure S10**). Similar nested topology and functional degree distributions can be observed in the PCNs generated with the IGC-based search (**Supplementary Figure S11**).”

Point 2.4: In the section, Protein-level FR outcompetes diversity indices in detecting microbiome responses to environmental factors: In lines 256-258, you mentioned that “In patients diagnosed with (IBD), nFR_p levels were significantly lower than that of the non-IBD individuals”. What was the cause, was it due to a loss of taxa or a loss of function?

Response: We thank Reviewer #2 for this critical comment. We believe that the observed nFR_p differences between IBD and non-IBD individuals is largely due to the loss of redundant functions. If we simply compare functional distances between highly abundant taxa, we found that d_{ij} between these genera were significantly affected. We show in **Fig. R11** (and in **Fig. 5** of the manuscript) that the d_{ij} distributions shifted rightwards in UC and CD samples, as well as in inflamed regions (**Fig. R11A** and **Fig. R11B**). If we perform statistical analysis on d_{ij} values between inflamed vs non-inflamed regions, we can observe that a great proportion of d_{ij} values have a significant increase (**Fig. R11C**). This compelling evidence suggests that redundant functions were lost between taxa in IBD inflammations.

Point 2.5: In the section, Alteration of between-proteome functional distances in diseases and compound-treated microbiome: D_{ij} is the functional distance between taxa i and taxa j (considering their proteome content). In line 303, you mention that D_{ij} values increased in inflammation. To me, it makes sense that if the distance increases, the redundancy decreases. Yet, in line 304, you write that there is a significant contribution of between proteome functional distance decrease to the overall decrease of nFR_p in IBD samples. Either the “proteome functional distance” in that sentence does not correspond to D_{ij} and then I’m a bit confused, or the “proteome functional distance” should increase and the sentence is wrong. Can you clarify please?

Response: We are sorry for causing such confusion. Indeed, the functional redundancy **decreases** if the functional distance **increases**. We apologize for the mistake in the sentence “suggesting a significant contribution of between-proteome functional distance **decrease** to the overall decrease of nFR_p in IBD samples”, where we meant to say “**increase**”. We have fixed this mistake in the revised manuscript (see page 12, line 341).

Point 2.6: Methodology section:

In line 609, you mentioned that Jensen-Shannon divergence and Kullback–Leibler divergence were calculated using the R package LaplacesDemon. However, in the results only KL was used. I actually had initially a random, out of curiosity question: What features of the D_{ij} distributions made you choose KL where people usually go for Jensen-Shannon? Were the results the same when using Jensen-Shannon?

Response: We thank Reviewer #2 for this comment. We apologize for our mistake. Taking Reviewer #2’s suggestion, we switched to using Jensen-Shannon divergence, which may be more broadly used and better accepted. We found that, together with the change of functional annotation method (from COG to KEGG-COG), the results are still much more consistent with our

previous comparisons (**Fig. R12**). We have switched the Kullback–Leibler divergence to the Jensen-Shannon divergence in the revised main text (page 12, lines 352-356).

Point 2.7: Supplementary figure section: In supplementary figure 3, you mention the UHGP database. I do not find it in the main text/in the methodology. Am I missing it?

Response: We thank Reviewer #2 for carefully examining the details of our manuscript. We previously also assessed the approach using the UHGP database with an LCA approach (**Figure R13**), and the findings agreed with the IGC-based approach. Considering that our computation of FR_p is robust and would not be affected by metaproteomics sample analysis or bioinformatic pipelines, other possible workflows leading to the successful construction of PCNs should also lead to same conclusions in terms of community ecology comparisons. As we have already conducted a thorough comparison of (1) GCN vs PCN (**Fig. 2I, 3A, S10, and S11**), (2) metagenome-based and IGC-based database search (e.g. **Fig. R8 & Fig. R9**), (3) three different functional annotation methods (**Supplementary Note 2**), and (4) four different metaproteomic strategies (**Supplementary Fig. S12**), we initially decided not to include the UHGP-based search to avoid unnecessary complexity. But we forgot to remove the UHGP-based search from Supplementary Figure 3. We thank Reviewer #2 for pointing this out. Considering Reviewer #2's next comment (Point 2.8), we present the UHGP-based workflow that we previously tested and compare its findings to that of the IGC-based workflow (**Figs. R13-17**). More details can be found in the responses to the following UHGP-related comments from Reviewer #2 (**Points 2.8 and 2.9**).

Point 2.8: In any case, in supplementary figure 3, the numbers of genera found by either UHGP or IGC are very different, but they are not for COG. Do you have an idea why?

Response: Yes, we believe it is related to the nature of UHGP database, we have added the details of the UHGP-based workflow below to help understand the reason:

The UHGP-based workflow: we searched the datasets using the UHGP V1.0 catalog (Almeida et al., *Nat Biotech*, 39:105-114, 2021). In contrast to the IGC database, the UHGP database is based on more than four thousand reference genomes of the human gut microbiome. This catalog does not remove redundant proteins and therefore the proteinGroups may contain multiple proteins from different taxa sharing certain groups of tryptic peptides. The protein ID of the UHGP V1.0 catalog has a genome identifier in the middle, which can be matched to the taxa directly from their provided taxonomic annotation generated by Genome Taxonomy Database Toolkit (GTDB-Tk).

For just a simple example, protein 'GUT_GENOME001575_00044' corresponds to GENOME001575, which is matched to s__*Faecalicatena faecis*. We therefore used a within-proteinGroup LCA approach to determine the taxonomic information across proteins in each of the proteinGroups (**Figure R13**). A special feature of the UHGP V1.0 database is that in its taxonomic lineage information, many bigger genera were sub-divided into different sub-genera (for example, *Bacteroides* proteins obtained from IGC results may correspond to g__*Bacteroides*,

g__*Bacteroides_A*, g__*Bacteroides_B* or g__*Bacteroides_C* proteins in the UGHP results), and due to this reason, the results showed a higher number of genus matches in the previous Supplementary Figure 3 in both GCN and PCN.

Point 2.9: And (going back to my first question) what is the effect of the database choice in the assessment of functional redundancy. Do your findings hold if you were to use the UHGP catalogue?

Response: In the previously submitted manuscript, we used the IGC-COG workflow which determines the taxonomic origination of proteins using the “protein-peptide bridge” method. This was based on a careful comparison of six different strategies (i.e., two bioinformatic workflows X three functional annotation approaches). We summarized what we previously did using a flow chart as **Figure R14**.

Since we’ve already stated the differences in functional annotation approaches and reasons in Supplementary Note 2, here we focus on comparing IGC-COC and UHGP-COG workflows. Although the UHGP-COG workflow is quite different from the IGC-COG workflow, we found that the two workflows yield highly similar network topologies (**Fig. R15**), consistent FR_p , and robust functional distance responses to environmental changes (**Fig. R16** and **Fig. R17**).

Note that although here in this response letter we presented our previous test results based on IGC-COC and UHGP-COG workflows, in the revised manuscript, we have changed the functional annotation to using KEGG KO complemented by COG annotations (i.e. KEGG-COG) after considering Reviewer #1’s comments. Also, we have switched to using Jensen-Shannon divergence instead of Kullback–Leibler divergence in **Fig. R18**. The fact that all these findings were not affected (as shown in **Fig. R2** and **R12**) clearly demonstrated that our metric of the protein-level function redundancy FR_p is quite **robust** and **generally applicable** to database search workflows based on different protein or gene catalogs.

Point 2.10: Minor comments/typos:

Line 248: SISPRORT  SISPROT

Response: We have corrected this typo (see main text, page 10, line 292).

Point 2.11: Line 328: Functionals → functions

Response: We have corrected this typo (see main text, page 13, line 371).

Point 2.12: Line 334: tend to be playing  tend to play

Response: We have revised that sentence accordingly (see main text, page 13, lines 376-377).

Point 2.13: Line 514-515: searching against the IGC or database What does that mean: OR database?

Response: We have removed 'or' (see main text, page 18, line 532).

Point 2.14: Figure 2I- Legend: There are not 4 rows. Maybe upper left panel, lower right panel,etc,.. would be clearer.

Response: We have revised the legend to use "upper left panel, lower right panel, etc" to point to the figures (see main text, page 30, lines 880-883).

Finally, we thank Reviewer #2 again for reviewing our manuscript and her/his very insightful comments that help us significantly improve the quality of our work. We hope our responses above have addressed her/his comments in a satisfactory manner.

Figures

Figure R1. Histogram of matching all COGs to KEGG KOs.

Figure R2. Comparisons between different functional annotation strategies. A-D. Comparison of different functional annotation methods in sensitivity of detecting functional distances variations: **A.** COG annotation showed significant alteration of between-genera d_{ij} distributions in response to drug treatments (J-S divergence). **B-C.** Using KEGG and CAZymes, despite the observation of similar patterns in the p-value heatmap, we did not observe any significant difference in J-S divergence between drugs (no asterisks shown). **D.** The KEGG-COG annotation method showed significant alteration of between-genera d_{ij} distributions in response to drug treatments, in agreement with panel (A), COG-based annotations. Asterisks indicate statistical significance at the 0.05 level (FDR adjusted p value, Pairwise Wilcoxon test). **E-F.** Comparison of functional redundancy and related metrics with the IBD dataset showed agreements between the COG and KEGG-COG based annotations.

Figure R3. Comparison of nFR_p values among different functional annotation methods.

Figure R4. KEGG-COG annotation induced slight decreases of nFR_p in the IBD dataset.

Figure R5. *In silico* community demonstrates the sensitivity of the nFR_p metrics. A. Illustration of experimental workflow. **B.** Genome-level FR, nFR, TD and FD of different *in silico* communities. **C.** Proteome-level FR, nFR, TD and FD of different *in silico* communities. **D.** Proteome-level FR, nFR, TD and FD of different *in silico* metaproteomes generated using proteomes cultured in different media.

Bacteroides stercoris	200	YYLFLK	TTYS	DIR	MVGAPPSSIGK	FGADTDNWMWPR	HTGDFSLFR	IYAGK	236
Bacteroides fragilis	201	YYLFLK	TVYNDIR		MVGAPPSSIGK	FGADTDNWMWPR	HTGDFSLFR	IYADK	237
LC-MS/MS identified peptide:					MVGAPPSSIGK	FGADTDNWMWPR	HTGDFSLFR		

Figure R6. An illustration showing why it is difficult to construct a species-level FR based on current metaproteomic techniques. Cyan, green and yellow-highlighted peptide sequences were identified by LC-MS/MS and those sequences could not be used to distinguish between the two species.

Figure R7. Comparison of nFR_p between rifaximin (RFXM) treatment vs DMSO groups in the RapidAIM dataset.

Figure R8. Comparison of FR, nFR, TD and FD between GCNs and PCNs generated using different bioinformatic workflows.

Figure R9. Network randomizations performed using the metagenomic database-based PCNs highly reproduced the findings from IGC database-based PCNs.

Figure R10. FR, nFR, TD and FD in GCN and PCN of simulated communities. Here, d_{ij} was calculated based on 1 (presence) or 0 (absence) of each function.

Figure R11. Between-proteome functional distances in IBD microbiomes. A. Distribution of d_{ij} values by diagnosis. **B.** Distribution of d_{ij} values by inflammation. **C.** Volcano plot showing altered d_{ij} values between inflamed and non-inflamed sampling sites. The results were based on microbial genera of the top 95% of overall protein biomass in the dataset.

Figure R12. Switching from K-L divergence to J-S divergence showed highly similar findings.

Figure R13. Alternative approaches can be used to construct the PCN. Here we show the workflow of constructing a pipeline based on the UHGP database.

Figure R14. Flow chart showing the comparison of different workflows and functional annotations for the PCN and FR analyses.

Acronyms in the figure:

PCN	Protein content network
FR	Functional redundancy
nFR	normalized Functional redundancy
UHGP	Unified Human Gastrointestinal Protein catalog
LCA	Lowest Common Ancestor approach
IGC	Integrated Gene Catalog
COG	Cluster of Orthologous Groups of proteins
KEGG	Kyoto Encyclopedia of Genes and Genomes
CAZyme	Carbohydrate-Active enZymes

Figure R15. Comparison of network topology and FR_p , nFR_p , FD_p , TD_p computed using IGC-COC and UHGP-COG workflows.

Figure R16. Comparison of nFR_p changes determined by different workflows in the RapidAIM dataset. Comparison between RFXM (rifaximin) treatment and DMSO (control) groups are shown.

Figure R17. K-L divergence between d_{ij} distributions in the control (DMSO) and that of the other compounds based on (A) the IGC-COG workflow, and (B) the UHGP-COG workflow.

REVIEWERS' COMMENTS

Reviewer #1 (Remarks to the Author):

The authors did a thorough job revising the paper, which looks much better now.

Reviewer #2 (Remarks to the Author):

I would like to thank the authors for the extra work they put in the manuscript. All my comments and questions have been thoroughly addressed and the manuscript was properly edited. I do not have further comments or requests.